# Focusing Where Vision Matters: Selective Training for Large Vision Language Models via Visual Information Gain

**Seulbi Lee** [1][†]  **Sangheum Hwang** [1]

## Abstract

Large Vision Language Models (LVLMs) have achieved remarkable progress, yet they often suffer from language bias, producing answers without relying on visual evidence. While prior work attempts to mitigate this issue through decoding strategies, architectural modifications, or curated instruction data, they typically lack a quantitative measure of how much individual training samples or tokens actually benefit from the image. In this work, we introduce Visual Information Gain (VIG), a perplexity-based metric that measures the reduction in prediction uncertainty provided by visual input. VIG enables fine-grained analysis at both sample and token levels, effectively highlighting visually grounded elements such as colors, spatial relations, and attributes. Leveraging this, we propose a VIG-guided selective training scheme that prioritizes high-VIG samples and tokens. This approach improves visual grounding and mitigates language bias, achieving superior performance with significantly reduced supervision by focusing exclusively on visually informative samples and tokens.

## 1. Introduction

Large Vision Language Models (LVLMs) (Liu et al., 2023; 2024b; Li et al., 2024; Gao et al., 2024; Chen et al., 2024; 2025c;a; Zhu et al., 2023; Chen et al., 2023) have demonstrated remarkable capabilities across a wide spectrum of multimodal tasks, ranging from image captioning (Ye et al., 2024; Peng et al., 2023) and visual question answering (Dai et al., 2023; Alayrac et al., 2022; Liu et al., 2024b) to more complex instruction following (Li et al., 2025; Lu

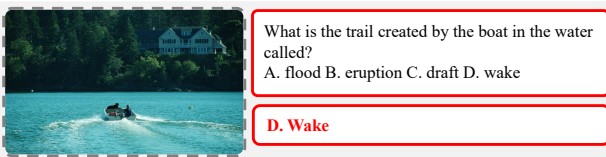

*(a)* A sample that can be answered from common sense

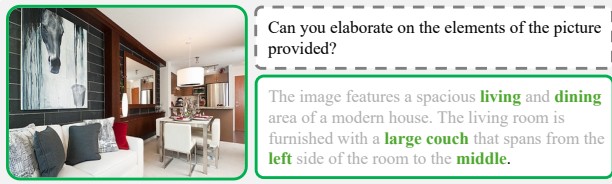

*(b)* A sample that requires fine-grained visual understanding

*Figure 1.* **Examples of LLaVA-1.5 instruction tuning data.** The dataset includes both samples and tokens with very different levels of visual dependency: some questions can be answered without looking at the image, whereas others need fine-grained visual details (highlighted in green).

et al., 2024; Wang et al., 2024) and reasoning (Wu et al., 2024; Bai et al., 2025). By combining powerful large language models (Touvron et al., 2023; Chiang et al., 2023; Bai et al., 2023; Young et al., 2024) with pre-trained vision encoders (Radford et al., 2021; Zhai et al., 2023), LVLMs can generate fluent and context-aware responses conditioned on both images and textual queries. However, despite this progress, it remains challenging to ensure that LVLMs are reliably grounded in the visual input rather than dominated by textual priors (Leng et al., 2024; Zhao et al., 2025; Liu et al., 2025a).

A growing body of work has shown that LVLMs often exhibit *language bias*: an over-reliance on language even when relevant visual evidence is available (Zhao et al., 2024). This bias manifests as *visual ignorance*, where the model effectively behaves as a text-only model and ignores salient image content (Liu et al., 2025b; Wan et al., 2025). It also leads to *hallucinations* (Goyal et al., 2017; Rohrbach et al., 2018; Liu et al., 2024a; Li et al., 2023; Gunjal et al., 2024), in which the model confidently describes objects or attributes that are not present in the image. Such behaviors call into question the reliability of LVLMs: to what extent do these models actually use the image, as opposed to merely being conditioned on it?

[†]Currently with Tomocube, Daejeon, Republic of Korea. [1]Department of Data Science, Seoul National University of Science and Technology, Seoul, Republic of Korea. Correspondence to: Sangheum Hwang <shwang@seoultech.ac.kr>.

*Proceedings of the 43rd International Conference on Machine Learning*, Seoul, South Korea. PMLR 306, 2026. Copyright 2026 by the author(s).

To mitigate language bias, prior work has mainly focused on model-level interventions. Training-free methods, such as contrastive decoding (Leng et al., 2024; Zhang et al., 2025), compare outputs with and without visual input at inference time, while other approaches boost image attention or modify attention mechanisms to encourage stronger visual grounding (Liu et al., 2025a; Jiang et al., 2025). In parallel, data-centric efforts construct higher-quality multimodal instruction datasets by leveraging stronger models or careful filtering (Chen et al., 2025a; Liu et al., 2024a; Yue et al., 2024). However, these approaches share a common limitation: they do not explicitly quantify, within a given multimodal dataset, how much each sample or token actually depends on visual information.

In practice, multimodal instruction-tuning datasets contain a heterogeneous mixture of examples: some can be answered from common sense or linguistic context alone (Fig. 1a), while others genuinely require fine-grained visual understanding (Fig. 1b). Although this heterogeneity appears not only at the sample level but also at the token level, visually grounded tokens (e.g., color, object attributes) are generally optimized under the same objective as non-visual or structural tokens (e.g., articles, auxiliaries, and discourse markers). When all such samples and tokens are treated with equal importance during training, the model receives no incentive to distinguish visually dependent signals from text-only patterns, and thus fails to fully acquire robust visual grounding, instead defaulting to easily exploitable linguistic shortcuts.

In this work, we take a data-centric perspective on language bias and visual grounding in LVLMs. We hypothesize that a key driver of language bias is the prevalence of weakly grounded, text-dominant examples in multimodal training datasets, combined with the uniform treatment of all tokens during training. We therefore ask a question: *can we directly measure how much each training sample and token benefits from the visual input, and use this signal to focus learning on genuinely visual evidence?* To this end, we introduce *Visual Information Gain (VIG)*, a perplexity-based metric that quantifies the contribution of visual information and supports both analysis and training of LVLMs.

The contributions of this work are threefold:

- We introduce *Visual Information Gain (VIG)*, a perplexity-based metric that quantifies the contribution of visual input by measuring the reduction in model uncertainty. VIG provides a model-agnostic and decomposable measure, enabling fine-grained analysis of visual dependency at both sample and token levels.

- We empirically demonstrate that VIG serves as a reliable indicator of visual grounding. Our analysis shows that VIG aligns with benchmark-level modality dependen-

cies and successfully identifies visually grounded tokens (such as colors, spatial relations, and attributes) while distinguishing them from tokens driven primarily by textual priors.

- We propose a VIG-guided selective training scheme that prioritizes high-VIG samples and tokens. This approach enhances data efficiency by pruning weakly grounded samples and focusing optimization on visually informative tokens. Notably, this strategy improves visual grounding and mitigates language bias, achieving superior performance with highly sparse supervision compared to full-data training.

## 2. Related Work

Despite the remarkable progress of LVLMs (Gao et al., 2024; Li et al., 2025; McKinzie et al., 2024; Gemini Team, 2023; Wang et al., 2024; OpenAI, 2023), recent studies have identified a persistent challenge known as language bias (Leng et al., 2024; Liu et al., 2025a; Zhao et al., 2025; Wan et al., 2025; Liu et al., 2025b). This refers to the tendency of LVLMs to produce visually ungrounded responses by over-relying on textual priors. Such bias often arises from language shortcuts (Niu et al., 2021; Vo et al., 2025) earned from noisy multimodal datasets, which are frequently synthesized from text-only LLMs and contain visually irrelevant or misleading captions (Chen et al., 2025a; Liu et al., 2024a; Yue et al., 2024). During training, LVLMs may find it statistically advantageous to exploit these textual patterns rather than attending to images. Empirical analyses further confirm that attention distributions within LVLMs tend to concentrate on textual tokens over visual features (Kaduri et al., 2025; Chen et al., 2025b), thereby limiting the model's active reference to visual information.

Efforts to generate visually grounded responses in LVLMs have primarily focused on reducing the dominance of textual priors or encouraging more effective visual grounding. Training-free approaches such as contrastive decoding (Leng et al., 2024; Zhang et al., 2025) attempt to compare model predictions with and without visual input, thereby suppressing language-driven responses during inference. However, these methods mainly circumvent rather than resolve the underlying issue and often incur additional inference overhead, as they do not modify how visual information is represented or utilized. Another line of work enhances visual grounding by boosting image attention (Liu et al., 2025a; Jiang et al., 2025), though its effects are often overly broad, amplifying irrelevant regions and occasionally introducing noise (Kang et al., 2025). Training-based strategies aim to address the issue more fundamentally. For instance, Zhao et al. (2025) proposes dual attention and soft-image guidance to explicitly promote visual utilization, though such techniques require architectural modifica-

tions. Collectively, these approaches highlight the need for bias mitigation strategies that can enhance visual grounding effectively while maintaining architectural simplicity and avoiding additional inference overhead.

## 3. Visual Information Gain

### 3.1. Preliminary

The prevalent architecture of LVLMs (Liu et al., 2023; 2024b; Li et al., 2024; Gao et al., 2024; Chen et al., 2025a) consists of three components: a pre-trained vision encoder $\mathcal{E}_v$, an adapter $\mathcal{P}$ and a pre-trained language model $\mathcal{D}$. Training typically follows a two-stage paradigm.

In the pre-training stage, the adapter $\mathcal{P}$ is trained on large-scale image–caption pairs formatted as single-turn instructions. For each image $I$ and its associated caption, a simple question $Q$ (e.g., "Describe this image") is randomly sampled to request a brief description, and the original caption serves as the target answer $A$. This process aligns the visual feature space with the semantic space of the language model, while keeping both $\mathcal{E}_v$ and $\mathcal{D}$ frozen. Subsequently, in the instruction tuning stage, the model is fine-tuned on complex multimodal instruction-following data $(I, Q, A)$. In this stage, $Q$ represents a diverse, task-oriented question, and $A$ is the corresponding answer. This stage jointly optimizes $\mathcal{P}$ and $\mathcal{D}$ to enhance the capability of the model in multimodal reasoning and instruction following. For each sample, the visual feature and its projected embedding are obtained as $f_v = \mathcal{E}_v(I)$, $z_v = \mathcal{P}(f_v)$. The model's predictive distribution over answer tokens is denoted as $q_\theta(\cdot \mid a_{<t}, Q, z_v)$, parameterized by $\theta$. The per-sample instruction tuning objective is thus defined as:

$$\mathcal{L}(A \mid Q, I; \theta) = -\frac{1}{T} \sum_{t=1}^{T} \log q_\theta(a_t \mid a_{<t}, Q, z_v) \quad (1)$$

where $a_t$ denotes the $t$-th token in the answer $A$ and $T$ is the sequence length. For notational simplicity, we omit $\theta$ and denote the model's predictive distributions under different conditioning as $q_Q(\cdot) = q_\theta(\cdot \mid Q)$ and $q_{I,Q}(\cdot) = q_\theta(\cdot \mid I, Q)$ which correspond to predictions without and with visual input, respectively.

### 3.2. Definition of VIG

To measure the sample-level contribution of visual information, we introduce VIG, which quantifies how much the inclusion of image $I$ reduces the model's uncertainty in predicting the answer $A$ given the question $Q$.

Formally, we define VIG as the log-ratio between the model's perplexities (PPL) on the same answer $A$ with and without visual conditioning:

$$\text{VIG} = \log\left(\frac{\text{PPL}(A \mid Q)}{\text{PPL}(A \mid Q, I)}\right) \quad (2)$$

where $\text{PPL}(A \mid Q)$ and $\text{PPL}(A \mid Q, I)$ denote the perplexities evaluated under the predictive distributions of the model $q_Q$ and $q_{I,Q}$, respectively. To simulate the absence of visual information within the LVLM architecture, we calculate $\text{PPL}(A|Q)$ using a blurred image that removes visual cues, as proposed by Xing et al. (2025) (see Appendix C for details). A higher VIG value indicates that the model's uncertainty is substantially reduced when visual information is provided, implying that the image plays a critical role in producing the correct answer.

To establish a theoretical foundation for VIG, we reformulate it in terms of cross-entropy loss and KL divergence. Using the relationship $\text{PPL} = \exp(\mathcal{L})$, where $\mathcal{L}$ is the cross-entropy loss, Eq. 2 can be rewritten as:

$$\text{VIG} = \mathcal{L}(A|Q) - \mathcal{L}(A|Q, I). \quad (3)$$

This formulation shows that VIG represents the reduction in cross-entropy loss attributable to the inclusion of the visual input. The cross-entropy between a ground-truth distribution $p$ and the model's predictive distribution $q$ is $\mathcal{L}(p, q) = H(p) + D_{KL}(p||q)$ where $H(p)$ is the intrinsic entropy of the target distribution and $D_{KL}$ denotes the KL divergence. Then, VIG can be rewritten as:

$$\text{VIG} = [H(p_{A|Q}) - H(p_{A|I,Q})] \\ + [D_{\text{KL}}(p_{A|Q}\|q_Q) - D_{\text{KL}}(p_{A|I,Q}\|q_{I,Q})] \quad (4)$$

where $p_{A|Q}$ and $p_{A|I,Q}$ represent the true conditional answer distributions given text-only and multimodal inputs, respectively, while $q_Q$ and $q_{I,Q}$ are the corresponding model predictive distributions.

In general, incorporating $I$ reduces the intrinsic uncertainty of the true answer distribution, such that $H(p_{A|I,Q}) < H(p_{A|Q})$. However, under deterministic supervision (typical in VQA and captioning datasets), we adopt the empirical distribution determined by the single ground-truth answer as the target $p$. In this case, $p$ is a Dirac delta distribution (one-hot), so the intrinsic entropy terms vanish: $H(p_{A|Q}) = H(p_{A|I,Q}) = 0$. Thus, Eq. 4 simplifies to:

$$\text{VIG} = \left[D_{\text{KL}}(p_{A|Q}\|q_Q) - D_{\text{KL}}(p_{A|I,Q}\|q_{I,Q})\right]. \quad (5)$$

Consequently, VIG quantifies empirically how much the visual information reduces the divergence between the model's predictive distribution and the ground truth.

Expanding Eq. 3, VIG can be expressed as the average of token-wise loss differences over $A = (a_1, \ldots, a_T)$:

$$\text{VIG} = \frac{1}{T} \sum_{t=1}^{T} [-\log q_\theta(a_t \mid a_{<t}, Q)] \\ - [-\log q_\theta(a_t \mid a_{<t}, Q, z_v)] \quad (6)$$

*Table 1.* **VIG's sensitivity to the degree of the visual grounding.** Examples from the MS-COCO (Lin et al., 2014) validation set show that VIG quantitatively captures the strength of visual support: high positive for a perfect match, moderate positive for partial grounding, and negative for a conflicting image.

| Question | What do you see on the floor near the red towel? |
|---|---|
| Answer | A white cat sitting on the floor next to his bowl. |
| Image | 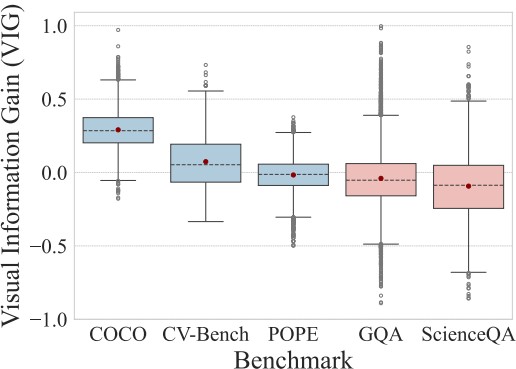 |
| VIG | 0.923    0.409    -0.520 |

where $z_v = \mathcal{P}(\mathcal{E}_v(I))$ denotes the visual embedding. Each term $-\log q_\theta(a_t \mid a_{<t}, Q)$ and $-\log q_\theta(a_t \mid a_{<t}, Q, z_v)$ represents the token-level cross-entropy loss computed without and with visual conditioning. This decomposition reveals that although VIG is defined at the sample level, it inherently reflects the aggregate contribution of per-token loss reductions. Analyzing token-level loss differences, therefore, provides a fine-grained view of which parts of a response strongly depend on visual information.

Throughout this work, we compute VIG using models after the *pre-training stage*, where the adapter $\mathcal{P}$ is trained to align the visual feature space with the language semantic space while keeping the vision encoder $\mathcal{E}_v$ and the language model $\mathcal{D}$ frozen. This ensures that the model has established basic visual–textual correspondence, allowing VIG to meaningfully reflect the contribution of visual information.

### 3.3. Analysis

To empirically validate the effectiveness of VIG, we conduct analyses based on LLaVA-v1.5 7B after the pre-training stage, where the model has learned to establish correspondences between visual and textual modalities.

**VIG is a fine-grained measure of visual grounding.** To isolate the impact of the visual input, we vary only the image while keeping the question–answer pair fixed. As shown in Tab. 1, VIG effectively quantifies the degree of visual support for a given textual description. The first image, perfectly aligned with the text, produces a high positive VIG of 0.923. The second image depicts the correct subject (`cat`) but mismatches an attribute (`white`), yielding a moderate positive VIG of 0.409. In contrast, the dog image contradicts the textual content, resulting in a negative VIG of $-0.520$. These examples confirm that VIG serves as a sensitive and reliable metric for measuring the extent to which visual information reduces model uncertainty. Additional qualitative examples are provided in Appendix A.

*Figure 2.* **VIG distribution across benchmarks.** Blue benchmarks (COCO, CV-Bench, POPE) show stronger multimodal interaction, while red benchmarks (GQA, SQA) exhibit weaker visual dependency.

*Table 2.* **Tokens and their loss differences in LLaVA-1.5 instruction-tuning data.** Red regions show large positive loss differences (strong visual grounding), while blue regions exhibit near-zero or negative differences (weak visual contribution).

| Red | white (3.59), black (6.08), lying (5.99), flying (5.73) sitting (4.22), standing (5.10), reading (5.974), crowd (3.30) |
|---|---|
| Blue | a (0.03), of (0.01), the (0.03), ize (0.00), which (0.00), are (-0.02), The (0.04), - (-1.10) |

**VIG aligns with benchmark-level modality dependency.** Previous studies have suggested that LVLM benchmarks differ in the extent to which they rely on visual versus textual information (Madaan et al., 2025): COCO (Lin et al., 2014), CV-Bench (Tong et al., 2024), and POPE (Li et al., 2023) generally require substantial visual understanding, whereas benchmarks such as GQA (Hudson & Manning, 2019) and SQA (Lu et al., 2022) are often considered to be more text-dominant. To examine whether VIG captures these tendencies, we compute the VIG score for every sample across benchmarks and visualize the distributions of sample-level VIG values in Fig. 2. We observe that COCO exhibits a distribution shifted toward positive values, consistent with the expectation that image captioning relies heavily on visual inputs. POPE and CV-Bench display a distribution centered near zero, suggesting a balanced dependency where models may utilize both visual evidence and textual cues. In contrast, GQA and SQA show distributions skewed toward negative values. This aligns with prior findings that these benchmarks often exhibit more reliance on textual information, where introducing visual information can inadvertently increase prediction uncertainty compared to text-only inference. Overall, these results suggest that VIG can characterize the sample-level modality dependency across LVLM benchmarks.

**VIG captures token-level visual grounding.** To empirically examine the token-level decomposition in Eq. 6, we

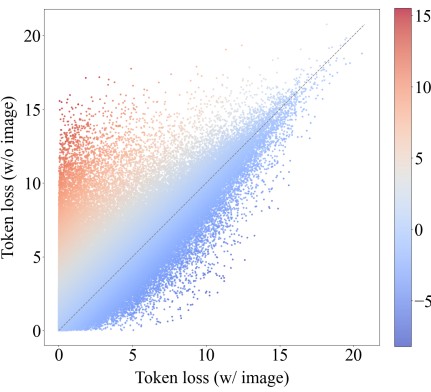

Figure 3. **Visualizing the token-level VIGs.** Each point shows a token's prediction loss with ($x$-axis) and without ($y$-axis) visual input. The color encodes the token-level loss difference ($y - x$).

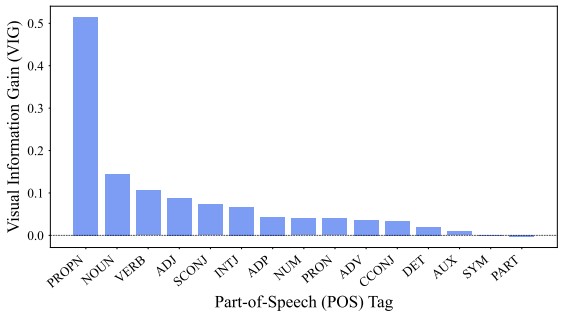

Figure 4. **Average VIG by POS category.** Visual relevant categories show higher VIG than function words, suggesting that VIG emphasizes linguistically informative tokens for visual grounding.

visualize the relationship between token losses with and without visual conditioning on the LLaVA-1.5 instruction-tuning dataset in Fig. 3. Points along the diagonal $y = x$ correspond to tokens unaffected by visual input. Notably, the scatter reveals a broad spectrum of visual dependency at the token level. Tokens with high VIG (visualized in red) appear above the diagonal, indicating that prediction losses are substantially reduced by the image. As detailed in Tab. 2, these tokens largely correspond to visually salient concepts such as colors, spatial relations, and physical states, i.e., elements that are often unpredictable from text alone. Conversely, tokens with low or negative VIG (blue region), such as articles and prepositions, primarily serve syntactic purposes. Furthermore, we perform a corpus-level part-of-speech (POS) analysis on the dataset. As shown in Fig. 4, the results reveal a clear pattern: visually grounded words (e.g., proper nouns, nouns, and verbs) consistently exhibit higher positive VIG scores, whereas function tokens (e.g., particles, symbols) cluster near zero. These findings confirm that sample-level visual dependency measured by VIG arises from the cumulative contribution of such visually grounded tokens.

### 3.4. VIG-Guided Selective Training

To demonstrate the practical utility of VIG, we adopt the principle of selective modeling, recently shown to be effective for LLMs (Lin et al., 2024). For the $i$-th training sample $(I_i, Q_i, A_i)$ with answer length $T_i$, we denote its sample-level VIG by $\mathrm{VIG}_i$ and its token-level visual gain by $\mathrm{VIG}_{i,t}$, where $\mathrm{VIG}_{i,t}$ corresponds to the token-wise loss difference term in Eq. 6. Then Eq. 6 can be rewritten as

$$\mathrm{VIG}_i = \frac{1}{T_i} \sum_{t=1}^{T_i} \mathrm{VIG}_{i,t}. \tag{7}$$

We use these quantities to perform VIG-guided selective training. First, we rank all training samples by $\mathrm{VIG}_i$ and select the top $p\%$. Let $\tau_p$ denote the corresponding threshold, i.e., the minimum $\mathrm{VIG}_i$ within this top-$p\%$ set, and define

$$\mathcal{S}_p = \{ i \mid \mathrm{VIG}_i \geq \tau_p \} \tag{8}$$

as the index set of selected samples. This sample-level filtering allows the model to focus on examples that provide substantial visual gains and reduces the influence of weakly grounded, text-dominant data. Within this curated subset, we further perform token-level selection using the same threshold. For each $i \in \mathcal{S}_p$, we define the set of visually informative tokens as

$$\mathcal{T}_i^+ = \{ t \mid \mathrm{VIG}_{i,t} \geq \tau_p \}, \tag{9}$$

and compute the loss only on these tokens. Concretely, during instruction tuning, we feed the full answer sequence $A_i$ and compute logits at every timestep, but the loss is computed exclusively over tokens in $\bigcup_{i \in \mathcal{S}_p} \mathcal{T}_i^+$, ensuring that unselected tokens do not contribute to gradient updates. Reusing the same threshold $\tau_p$ at both the sample and token levels is an intentional design choice to avoid introducing additional hyperparameters, and it concentrates optimization on the most visually informative regions of the data, leading to more visually grounded and data-efficient learning.

## 4. Experiment

### 4.1. Tasks and Benchmarks

We evaluate the effectiveness of VIG-based selective training on two categories of benchmarks: vision understanding and hallucination evaluation. For vision understanding, which spans basic recognition to more complex multimodal reasoning, we use LLaVA$^\mathrm{W}$ (Liu et al., 2023), MMVet (Weihao et al., 2024), MMBench (Yuan et al., 2024), CV-Bench (Tong et al., 2024), and DocVQA (Mathew et al., 2021). To assess hallucination behavior, we adopt POPE (Li et al., 2023), CHAIR (Rohrbach et al., 2018), and MMHal (Sun et al., 2024). Further details are provided in the Appendix B.

## 4.2. Overall Performance and Data Efficiency

**Setup.** We evaluate VIG-guided selective training on LLaVA-1.5 7B, LLaVA-1.5 13B (Liu et al., 2024b), ShareGPT4V 7B (Chen et al., 2025a), and Open-Qwen2VL 2B (Wang et al., 2025). For LLaVA-1.5 and ShareGPT4V, following Sec. 3, we first obtain aligned LVLMs by performing alignment training on image–caption data (558K pairs for LLaVA-1.5 and 1.2M pairs for ShareGPT4V). The subsequent instruction-tuning datasets for these models contain roughly 665K instances each: about 40K of these are text-only, and the remaining $\sim$625K are multimodal (image–instruction–answer) samples. For Open-Qwen2VL 2B, we utilize its official pretrained checkpoint and perform instruction tuning on a 1M multimodal subset randomly sampled from the MAmmoTH-VL dataset (Guo et al., 2025). Across all training configurations, we leave any text-only samples unchanged and compute $\text{VIG}_i$ exclusively on the multimodal subsets using the aligned models.

For VIG-guided sample selection in the instruction-tuning stage, we set $p = 70$ and use the corresponding threshold $\tau_{70}$ for all experiments. We rank all multimodal instruction samples by $\text{VIG}_i$ and retain the top 70%, resulting in 437K multimodal instruction samples for LLaVA-1.5, 436K for ShareGPT4V, and 701K for Open-Qwen2VL, respectively. Within these selected samples, VIG-guided token selection adopts the same threshold: only tokens whose token-level gain $\text{VIG}_{i,t} \geq \tau_{70}$ are included in the loss computation during instruction tuning. Details on the training data and configurations are provided in the Appendix D.

**Results.** As shown in Tab. 3, VIG-based selective training yields strong data efficiency: *By training on only 70% of the samples and further pruning supervision via token-level masking, all models exceed their vanilla counterparts.*

For LLaVA-1.5 7B, VIG training optimizes on only 38.45M target tokens yet improves performance on all benchmarks. The effect is even more pronounced for LLaVA-1.5 13B: although it is optimized on only 12.14M tokens, it boosts performance across all benchmarks. It suggests that larger models can make more effective use of carefully selected, visually grounded data, even when trained on substantially fewer tokens. For ShareGPT4V 7B, the vanilla model already outperforms LLaVA-1.5 7B due to its stronger image–text alignment. On top of this, VIG selection (only 39.20M tokens) further improves performance with particularly clear gains on vision-understanding benchmarks.

To verify that VIG generalizes across diverse architectural designs and datasets, we also evaluate our method on Open-Qwen2VL 2B (Wang et al., 2025). As shown in Tab. 3, despite the significant differences in architecture and training data, VIG training reduces active tokens by 41% while consistently improving performance across all vision un-

derstanding and hallucination benchmarks. These results demonstrate that language bias remains a persistent challenge even in newer, large-scale datasets. This confirms that VIG serves as a robust and architecture-agnostic solution to effectively mitigate this bias.

Note that while we fix the sample selection ratio at $p = 70$ for all models, the resulting "# Active Tokens" differ because the distribution of token-level $\text{VIG}_{i,t}$ varies across models: applying the same threshold $\tau_{70}$ thus retains different proportions of tokens. Overall, these results demonstrate that prioritizing visually important data at both sample and token levels can substantially reduce the amount of supervision required, while even improving performance on both vision understanding and hallucination benchmarks. We further provide qualitative comparisons in Appendix D.3.

## 4.3. Comparison with Existing Methods

**Baselines.** We compare our approach on LLaVA-1.5 7B with four recent methods that aim to strengthen visual grounding. We include three training-free methods: VCD (Leng et al., 2024), which utilizes contrastive decoding to suppress language priors, PAI (Liu et al., 2025a), which explicitly amplifies visual attention, and VAR (Kang et al., 2025), which redistributes attention to mitigate attention sink issues. Also, we compare against LACING (Zhao et al., 2025), a training-based method that fine-tunes the model to enforce the usage of visual information.

**Results.** As shown in Tab. 4, existing methods exhibit notable trade-offs. LACING achieves strong results on MMVet, MMBench, and CHAIR but degrades on fine-grained document understanding (DocVQA). Similarly, VCD and PAI improve hallucination metrics but often at the expense of general vision understanding capabilities. In contrast, VAR offers a more balanced trade-off, serving as a strong inference-time baseline.

Our VIG-trained model achieves competitive or superior performance across all benchmarks without any architectural changes or inference-time overhead. Unlike LACING, which sacrifices performance in specific domains, VIG training strictly improves over the vanilla baseline on all vision-understanding tasks, including LLaVA$^W$, MMVet, MMBench, and DocVQA. Notably, on MMHal, VIG substantially reduces hallucination while simultaneously boosting the overall score, demonstrating that training exclusively on visually informative tokens effectively strengthens visual grounding.

Furthermore, VIG training exhibits strong orthogonality to existing approaches. As detailed in Tab. 4, integrating VIG with inference-time strategies (VCD, PAI, VAR) consistently yields additive gains in both vision understanding and hallucination robustness. It also combines naturally with

*Table 3.* **Quantitative comparison on vision understanding and hallucination benchmarks.** We report results for four LVLMs: LLaVA-1.5 7B/13B, ShareGPT4V 7B, and Open-Qwen2VL 2B. "VIG training" denotes our VIG-guided selective training. "# Sample Tokens" represents the total number of answer tokens contained in the multimodal samples retained after sample-level selection. "# Active Tokens" refers to the effective number of tokens that contribute to the loss computation after applying token-level masking. $\Delta$ indicates the percentage reduction in token count or the performance improvement compared to the vanilla baseline. For each metric, ↑ indicates higher is better and ↓ indicates lower is better. **Bold** indicates entries where VIG-guided selective training outperforms the vanilla model.

| Model | # Sample Tokens | # Active Tokens | Vision Understanding | | | | | POPE | | Hallucination CHAIR | | MMHal | |
|---|---|---|---|---|---|---|---|---|---|---|---|---|---|
| | | | LLaVA$^W$ | MMVet | MMBench | CV-Bench | DocVQA | F1 ↑ | Acc. ↑ | $C_S$ ↓ | $C_I$ ↓ | Score ↑ | Hall. ↓ |
| | | | Score ↑ | | | Acc. ↑ | | | | | | | |
| LLaVA-1.5 7B | 58.61M | 58.61M | 59.02 | 28.62 | 65.46 | 59.18 | 22.31 | 85.90 | 87.08 | 52.93 | 14.99 | 1.71 | 71.25 |
| + VIG training | 51.17M | 38.45M | 61.22 | 32.71 | 66.33 | 62.48 | 22.51 | 85.93 | 87.47 | 47.00 | 12.80 | 2.23 | 62.78 |
| $\Delta$ | **-13%** | **-34%** | **+2.20** | **+4.09** | **+0.87** | **+3.30** | **+0.20** | **+0.03** | **+0.39** | **+5.93** | **+2.19** | **+0.52** | **+8.47** |
| LLaVA-1.5 13B | 58.61M | 58.61M | 72.01 | 36.19 | 67.52 | 60.16 | 24.08 | 85.72 | 87.05 | 51.96 | 13.22 | 2.05 | 67.09 |
| + VIG training | 28.94M | 12.14M | 73.45 | 36.87 | 68.67 | 62.89 | 25.27 | 86.95 | 87.53 | 48.19 | 13.19 | 2.12 | 63.11 |
| $\Delta$ | **-51%** | **-79%** | **+1.44** | **+0.68** | **+1.15** | **+2.73** | **+1.19** | **+1.23** | **+0.48** | **+3.77** | **+0.03** | **+0.07** | **+3.98** |
| ShareGPT4V 7B | 60.33M | 60.33M | 64.49 | 33.16 | 65.89 | 60.19 | 26.15 | 85.69 | 86.98 | 28.12 | 7.88 | 1.80 | 70.99 |
| + VIG training | 49.34M | 39.20M | 66.67 | 35.51 | 67.81 | 63.90 | 28.23 | 87.15 | 87.24 | 25.66 | 6.56 | 2.01 | 66.12 |
| $\Delta$ | **-18%** | **-35%** | **+2.18** | **+2.35** | **+1.92** | **+3.71** | **+2.08** | **+1.46** | **+0.26** | **+2.46** | **+1.32** | **+0.21** | **+4.87** |
| Open-Qwen2VL 2B | 4.10B | 4.10B | 63.89 | 37.77 | 78.59 | 68.99 | 41.05 | 86.48 | 87.27 | 29.84 | 7.55 | 1.99 | 64.28 |
| + VIG training | 3.29B | 2.42B | 65.17 | 39.01 | 79.67 | 70.01 | 44.05 | 87.99 | 87.98 | 27.74 | 6.98 | 2.23 | 62.56 |
| $\Delta$ | **-20%** | **-41%** | **+1.28** | **+1.24** | **+1.08** | **+1.02** | **+3.00** | **+1.51** | **+0.71** | **+2.10** | **+0.57** | **+0.24** | **+1.72** |

*Table 4.* **Quantitative comparison with existing methods on LLaVA-1.5 7B.** We compare our approach against recent training-free (VCD (Leng et al., 2024), PAI (Liu et al., 2025a), VAR (Kang et al., 2025)) and training-based approaches (LACING (Zhao et al., 2025)), proposed to improve visual grounding. "VIG training" denotes VIG-guided selective training on LLaVA-1.5 7B. Values in parentheses indicate the performance improvement over the vanilla model. **Bold** indicates entries where VIG-guided selective training improves over the vanilla model.

| Model | Vision Understanding | | | | POPE | | Hallucination CHAIR | | MMHal | |
|---|---|---|---|---|---|---|---|---|---|---|
| | LLaVA$^W$ | MMVet | MMBench | DocVQA | F1 ↑ | Acc. ↑ | $C_S$ ↓ | $C_I$ ↓ | Score ↑ | Hall. ↓ |
| | Score ↑ | | | Acc. ↑ | | | | | | |
| LLaVA-1.5 7B | 59.02 | 28.62 | 65.46 | 22.31 | 85.90 | 87.08 | 52.93 | 14.99 | 1.71 | 71.25 |
| + VCD | 60.55 | 27.01 | 64.34 | 22.98 | 86.53 | 86.61 | 49.81 | 13.69 | 1.67 | 76.01 |
| + PAI | 57.54 | 27.99 | 65.45 | 21.98 | 85.98 | 85.84 | 35.46 | 12.11 | 1.78 | 72.31 |
| + VAR | 61.11 | 30.98 | 66.30 | 23.11 | 86.10 | 87.12 | 50.09 | 14.77 | 2.14 | 62.89 |
| + LACING | 61.09 | 34.15 | 66.45 | 21.45 | 85.58 | 86.68 | 30.85 | 11.73 | 2.12 | 64.54 |
| VIG training | 61.22 (**+2.20**) | 32.71 (**+4.09**) | 66.33 (**+0.87**) | 22.51 (**+0.20**) | 85.93 (**+0.03**) | 87.47 (**+0.39**) | 47.00 (**+5.93**) | 12.80 (**+2.19**) | 2.23 (**+0.52**) | 62.78 (**+8.47**) |
| + VCD | 61.98 (**+1.43**) | 32.98 (**+5.97**) | 67.65 (**+3.31**) | 23.15 (**+0.17**) | 86.11 (-0.42) | 87.12 (**+0.51**) | 44.99 (**+4.82**) | 12.00 (**+1.69**) | 2.11 (**+0.44**) | 62.77 (**+13.24**) |
| + PAI | 59.99 (**+2.45**) | 32.69 (**+4.70**) | 67.34 (**+1.89**) | 22.22 (**+0.24**) | 86.12 (**+0.14**) | 87.49 (**+1.65**) | 32.12 (**+3.34**) | 11.86 (**+0.25**) | 2.43 (**+0.65**) | 62.90 (**+9.41**) |
| + VAR | 63.00 (**+1.89**) | 34.91 (**+3.93**) | 67.66 (**+1.36**) | 23.22 (**+0.11**) | 86.99 (**+0.89**) | 87.50 (**+0.38**) | 44.98 (**+5.11**) | 11.87 (**+2.90**) | 2.54 (**+0.40**) | 58.90 (**+3.99**) |
| + LACING | 62.99 (**+1.90**) | 37.01 (**+2.86**) | 67.89 (**+1.45**) | 22.02 (**+0.57**) | 86.19 (**+0.61**) | 87.39 (**+0.71**) | 28.11 (**+2.74**) | 9.97 (**+1.76**) | 2.71 (**+0.59**) | 56.10 (**+8.44**) |

the training-based method LACING, since VIG operates at the data level while LACING modifies the architecture. Their combination ("VIG training + LACING") achieves the strongest overall performance on MMVet (37.01) and MMBench (67.89). These results confirm that VIG serves as a fundamental, data-centric enhancement that complements diverse visual grounding strategies.

### 4.4. Analysis

**Increased attention to visual tokens.** To better understand why VIG-guided training improves performance, we examine how the model allocates attention to visual tokens. Following the analysis protocol of Kaduri et al. (2025), we evaluate the models on a subset of the MS-COCO (Lin et al., 2014) validation set and measure the proportion of attention weights assigned to image tokens relative to the total attention at each layer. This ratio effectively summarizes the strength of visual reference across network depth. As shown in Fig. 5, the VIG-trained model consistently assigns a larger fraction of attention to visual tokens than the vanilla LLaVA-1.5 7B model. The gap is especially pronounced in the middle layers, which have been identified as crucial for semantic feature extraction from visual inputs (Jiang et al., 2025; Kaduri et al., 2025). These results indicate that VIG-guided selective training encourages the model to refer more strongly to visual evidence. Further analyses on LLaVA-1.5 13B and ShareGPT4V 7B are provided in the Appendix E.1.

**Reducing language bias via VIG-guided training.** Recent work has identified a form of text bias in LVLMs, termed "blind faith in text", where models follow misleading textual cues instead of relying on the image (Kembhavi et al., 2025). To evaluate whether VIG-guided training mitigates this behavior, we adopt the evaluation protocol of Kembhavi et al. (2025) on VQAv2 (Goyal et al., 2017), using LLaVA-1.5 7B as the baseline. In this setup, the model is presented with the original image together with a corrupted description, where spurious text is appended to encourage an incorrect answer. Fig. 6 reports the accuracy on clean inputs (Base),

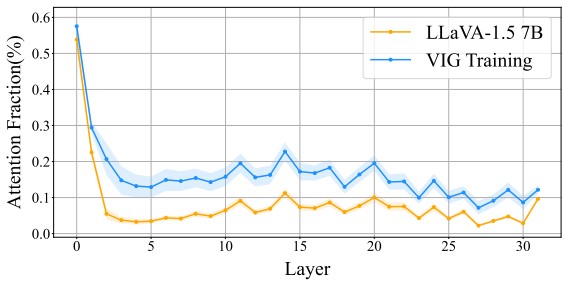

*Figure 5.* **Attention fraction allocated to visual tokens.** Compared to LLaVA-1.5 7B, VIG training assigns significantly more attention to visual tokens across all layers.

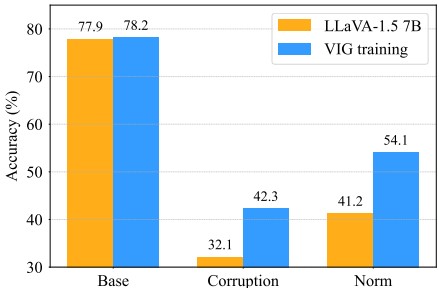

*Figure 6.* **Evaluation of text reliance under textual corruption.** Base: accuracy on clean inputs. Corruption: accuracy when the same image is paired with a corrupted caption containing a conflicting description. Norm: corruption accuracy normalized by the corresponding Base (Corruption/Base).

the accuracy under textual corruption (Corruption), and the normalized score (Norm=Corruption/Base). While both models achieve similar accuracy on clean questions, the VIG-trained model attains higher accuracy in the corruption setting and a larger normalized score, indicating that the VIG-trained model effectively resists textual interference by grounding its predictions in the visual input, even when misleading text is present. These results suggest that VIG-guided selective training reduces the model's reliance on spurious textual cues and encourages stronger use of visual evidence. Furthermore, as detailed in Appendix E.2, VIG yields substantially higher robustness against such textual corruption compared to inference-time methods like VCD, confirming that VIG addresses language bias fundamentally at the representational level. Appendix E.2 provides further analyses on LLaVA-1.5 13B and ShareGPT4V 7B.

**Preserving text comprehension.** We examine whether VIG-guided training affects the model's text-only capabilities by evaluating LLaVA-1.5 7B on standard text-only benchmarks. As shown in Tab. 5, VIG training maintains comparable performance on GSM8K (Cobbe et al., 2021), MMLU (Hendrycks et al., 2021), HellaSwag (Zellers et al., 2019), and TruthfulQA (Lin et al., 2022), with only marginal variations across all four benchmarks. This behavior is expected from the design of VIG-guided training: (1) VIG-

*Table 5.* **Comparison on text-only benchmark performance.** Performance on GSM8K, MMLU, HellaSwag, and TruthfulQA remains largely unchanged with VIG training, suggesting that improvements in visual grounding do not come at the expense of text comprehension capabilities.

| Model | GSM8K | MMLU | HellaSwag | TruthfulQA |
|---|---|---|---|---|
| LLaVA 1.5 7B | 15.33 | 51.68 | 76.09 | 45.86 |
| + VIG training | 14.99 | 51.35 | 76.11 | 46.01 |

*Table 6.* **Ablation study of selection levels on LLaVA-v1.5 7B.** "Random" trains on a random 70% subset of the data, "SS" selects the top 70% samples by VIG score (sample-level selection only), and "SS+TS" additionally applies token-level VIG selection. We report a single metric per benchmark: LLaVA$^W$ score, MMBench score, $C_S$ for CHAIR, and Hall. for MMHal.

| Model | LLaVA$^W$ ↑ | MMBench ↑ | CHAIR ↓ | MMHal ↓ |
|---|---|---|---|---|
| LLaVA1.5-7B | 59.02 | 65.46 | 52.93 | 71.25 |
| Random | 56.91 | 55.97 | 54.88 | 74.49 |
| SS | 58.12 | 57.56 | 50.23 | 68.14 |
| SS+TS | **61.19** | **66.33** | **49.10** | **61.82** |

based selection is applied exclusively to the multimodal instruction data, leaving text-only supervision unchanged, and (2) tokens not selected by VIG are masked out from the loss rather than removed from the input, allowing the model to preserve the full linguistic context during training. We provide additional text-only evaluation results for LLaVA-1.5 13B and ShareGPT4V 7B in Appendix E.3, where the same overall trend is observed.

**Impact of VIG-based filtering on data distribution.** A key concern in selective training is whether filtering disproportionately removes rare concepts. We analyze the retained 70% of the LLaVA-665K dataset and observe that the retention rates for COCO object categories are nearly uniform across frequency tiers (head, torso, and tail). This confirms that VIG filtering preserves long-tail concepts. Furthermore, VIG filtering improves the composition of the retained data by selectively preserving tasks that require dense visual grounding: OCR-VQA (98.4%) and TextVQA (92.6%) are retained at particularly high rates, whereas more text-dominant tasks such as GQA (44.2%) are filtered more heavily. These results indicate that VIG-based selection effectively reflects sample-level visual dependency rather than simply filtering out rare words, while preserving long-tail concepts and data diversity. Detailed settings and dataset statistics are provided in Appendix E.4.

### 4.5. Ablation Study

**Effectiveness of VIG-based selection.** To validate the effectiveness of our VIG-guided selection strategy, we conduct an ablation study on LLaVA-1.5 7B using three settings: (i) training on a random 70% subset of the full data (*Random*),

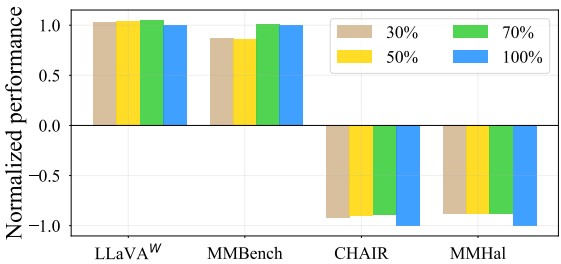

*Figure 7.* **Ablation study of selection ratio $p\%$ on LLaVA-1.5 7B.** We report a single metric per benchmark: LLaVA$^{\text{W}}$ score, MMBench score, $C_S$ for CHAIR, and Hall for MMHal. $p = 100$ corresponds to the vanilla model trained on the full instruction-tuning dataset (no VIG-based selection). All scores are normalized with respect to the $p = 100$ setting.

(ii) selecting the top 70% samples by VIG score without token-level filtering (*SS*), and (iii) applying both sample- and token-level selection (*SS+TS*).

As shown in Tab. 6, the *Random* setting yields slightly lower scores compared to the vanilla model. This marginal degradation aligns with prior observations that LLaVA retains around 95% of its performance even when trained on half of the instruction data (Liu et al., 2024b). In contrast, selecting the same number of samples based on VIG (*SS*) surpasses the *Random* setting and even outperforms the vanilla model across all four benchmarks, confirming that VIG is effective at identifying visually informative samples. Finally, incorporating fine-grained token selection (*SS+TS*) leads to the best results on every metric, highlighting that token-level filtering is crucial for maximizing the benefits of VIG-guided training. Please refer to Appendix F.1 for the detailed results.

**Effect of selection ratio $p$.** We investigate the sensitivity of VIG-guided training to the selection ratio $p$. Since $p$ determines the selection threshold $\tau_p$, a smaller $p$ implies a higher threshold, resulting in fewer selected samples and more aggressive token-level selection.

Fig. 7 reports normalized performance of LLaVA-1.5 7B for $p \in \{30, 50, 70, 100\}$, where $p = 100$ represents the vanilla model trained on the full instruction-tuning dataset. Note that at $p = 30$, $p = 50$, and $p = 70$, the model is updated with approximately 5%, 17%, and 65% of the total tokens, respectively. Overall, the impact of $p$ varies across benchmarks. On LLaVA$^{\text{W}}$, all reduced-ratio settings ($p < 100$) outperform the full-data baseline. Even the most aggressive setting ($p = 30$) yields strong results, suggesting that open-ended generation prioritizes data quality over quantity. In contrast, on MMBench, aggressive filtering ($p = 30, 50$) results in a slight performance drop, whereas $p = 70$ matches or exceeds the baseline. This implies that multiple-choice reasoning requires broader data coverage to maintain robustness across diverse topics. For halluci-

nation benchmarks (CHAIR and MMHal), all VIG-trained models ($p < 100$) consistently outperform the baseline. However, the performance gap between $p = 30$, 50, and 70 is marginal. These results demonstrate that VIG-based selection can substantially reduce the supervision cost while maintaining and often improving performance across a wide range of selection ratios. Detailed results are provided in the Appendix F.2.

## 5. Conclusion

We introduce *Visual Information Gain* (VIG), a perplexity-based metric that quantifies how much each multimodal training sample and token benefits from visual input. Our analysis shows that VIG correlates well with benchmark-level modality dependency and highlights visually grounded tokens such as colors, spatial relations, and object attributes, while deemphasizing tokens that can be predicted from text alone. Building on this, we propose a VIG-guided selective training scheme that prioritizes high-VIG samples and tokens, enabling LVLMs to achieve better vision understanding and hallucination robustness while utilizing only a fraction of the original supervision. We further demonstrate that VIG-based data selection is complementary to existing visual grounding strategies, yielding additional gains when combined. Overall, our results suggest that explicitly quantifying the visual contribution of training data is a promising direction for building LVLMs that more reliably use what they see.

A practical limitation of our approach is the computational overhead of computing VIG: for each multimodal instruction, we require additional forward passes using the aligned model. However, VIG scoring is a one-time, forward-only, and fully parallelizable procedure, and the resulting scores can be reused across training runs and model variants. Thus, our primary focus in this work is on maximizing *data efficiency*, reducing the amount of multimodal supervision needed to train a strong LVLM, rather than minimizing the overall computational cost. In addition, our empirical study is primarily demonstrated on the LLaVA-1.5 and ShareGPT4V families. Applying VIG-guided selection to other architectures and domains remains an important direction for future work.

## Acknowledgments

This work was supported by the National Research Foundation of Korea (NRF) under Grant [RS-2024-00352184] and [RS-2024-00354675] funded by the Ministry of Science and ICT (MSIT).

## Impact Statement

This work aims to improve the visual grounding of large vision-language models by prioritizing training samples and tokens that benefit most from visual input, thereby reducing reliance on textual priors and mitigating hallucinations. This may contribute to more reliable multimodal systems in applications such as visual assistance and document understanding, where faithful use of visual evidence is important. However, VIG-guided training is not designed to eliminate broader reliability and safety concerns, including biased or incomplete training data, uneven performance across domains or user groups, and overconfidence in high-stakes settings. Careful evaluation remains necessary before deploying such models in sensitive real-world applications.

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

# A. Additional Qualitative Examples of VIG

We provide additional qualitative examples in Tab. 7. Samples with high VIG scores exhibit a strong dependency on fine-grained visual evidence. In contrast, low or negative VIG samples contain questions that can be mostly answered through textual priors or common sense, validating VIG as a sensitive and reliable metric for measuring token- and sample-level visual grounding.

# B. Details of Benchmarks

**Visual understanding task.** To assess the model's capabilities in general visual perception and reasoning, we employ the following benchmarks.

- **LLaVA$^{\mathbf{W}}$** (Liu et al., 2023): LLaVA-Bench (In-the-Wild) comprises 24 images and 60 associated questions. It encompasses a diverse array of visual domains such as indoor and outdoor environments, memes, paintings, and sketches. The dataset is designed to assess the LVLMs' capability in handling complex tasks and generalizing to unfamiliar environments. It is evaluated by GPT-4 (`gpt-4o-2024-11-20`).

- **MMVet** (Weihao et al., 2024): MM-Vet evaluates the integrated capabilities of LVLMs in visual conversation across a broad range of tasks. Comprising 200 images and 218 questions with ground-truth references, it employs a GPT-4 (`gpt-4-0613`) evaluation framework to assess both the precision and utility of the model's responses.

- **MMBench** (Yuan et al., 2024): MMBench is a comprehensive benchmark containing roughly 3,000 multiple-choice questions that cover 20 skills. It uses GPT-3.5 (`gpt-3.5-turbo-0613`) to extract the final prediction label (A, B, C, D) from the model's response. Our evaluation focuses specifically on the English subset of the dataset. We report the results using the development split of the dataset.

- **CV-Bench** (Tong et al., 2024): Designed to assess foundational visual capabilities within a multimodal framework, CV-Bench comprises 2,638 manually verified instances derived from standard vision datasets such as ADE20k, COCO, and OMNI3D. By converting rich ground-truth annotations into natural language queries, it evaluates fundamental 2D (e.g., object counting, spatial relations) and 3D (e.g., depth ordering, relative distance) understanding. The benchmark is formatted as a multiple-choice question answering (MCQA) task. In our evaluation, we report the combined accuracy across all tasks to measure the model's overall visual reasoning.

- **DocVQA** (Mathew et al., 2021): DocVQA targets the task of visual document understanding, challenging models to extract and reason about information embedded in document images such as forms, invoices, and reports. In this study, we perform all evaluations using the official validation split and report accuracy.

**Hallucination evaluation task.** We evaluate the model's robustness to hallucination across the following benchmarks.

- **POPE** (Li et al., 2023): Polling-based Object Probing Evaluation (POPE) serves as a robust metric for assessing object hallucination. Constructed from MSCOCO (Lin et al., 2014), A-OKVQA (Schwenk et al., 2022), and GQA (Hudson & Manning, 2019), it comprises 27,000 query-answer pairs derived from 500 images per dataset. The core mechanism involves querying LVLMs about the existence of specific objects, with a balanced 50:50 ratio of existent to non-existent objects. To rigorously test the model, POPE employs three distinct negative sampling strategies: *random* (arbitrary missing objects), *popular* (high-frequency missing objects), and *adversarial* (co-occurring but absent objects). With six questions assigned per image, performance is evaluated using Accuracy, Precision, Recall, and F1 score. We report the average Accuracy and F1 score computed across all three negative sampling strategies.

- **CHAIR** (Rohrbach et al., 2018): To quantify object hallucination in generated captions, we employ the CHAIR (Captioning Hallucination Assessment with Image Relevance) metric. This method evaluates caption faithfulness by cross-referencing generated objects with those actually present in the image. The evaluation consists of two distinct metrics: $\text{CHAIR}_I$ (instance-level), which measures the proportion of hallucinated objects among all generated objects, and $\text{CHAIR}_S$ (sentence-level), which represents the percentage of captions containing at least one hallucination. The metrics are expressed by the following equations:

$$\text{CHAIR}_I = \frac{\#\text{hallucinated objects}}{\#\text{generated objects}}, \qquad \text{CHAIR}_S = \frac{\#\text{hallucinated captions}}{\#\text{generated captions}}. \tag{10}$$

*Table 7.* **Additional qualitative examples of VIG.** Examples from the MS-COCO (Lin et al., 2014) validation set show that VIG captures the strength of visual grounding: high positive for a strong match, near-zero for weak or partial grounding, and negative for a conflicting image. The value shown below each image is its VIG score.

| | | Strong grounding | Weak / partial grounding | Conflicting image |
|---|---|---|---|---|
| **Example 1** | Q. What is the name of the boat?
A. The answer is stress free. | | | |
| | | 1.331 | -0.074 | -0.530 |
| **Example 2** | Q. What Avenue is the stop sign on?
A. The stop sign say Washington. | | | |
| | | 0.887 | 0.031 | -0.425 |
| **Example 3** | Q. What is on the pizza?
A. In this image, what is shown on it is broccoli. | | | |
| | | 0.988 | -0.0487 | -0.6071 |
| **Example 4** | Q. What color is the person's shirt?
A. He is wearing red shirt. | | | |
| | | 0.909 | -0.016 | -0.694 |
| **Example 5** | Q. What image is on the top left of the bus marquee?
A. There is an image of people on the top left of the bus. | | | |
| | | 0.771 | -0.020 | -0.976 |

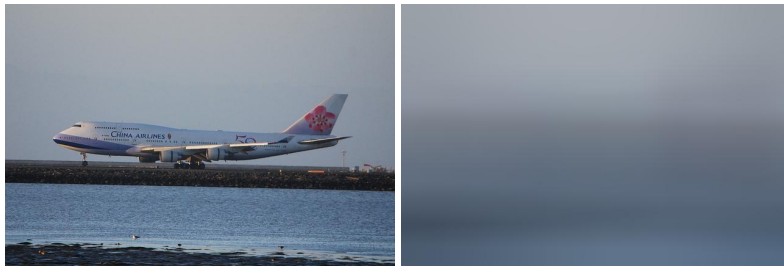

*(a)* Ground truth image.          *(b)* Image-absent condition.

*Figure C.1.* Examples of with and without image setting.

- **MMHal** (Sun et al., 2024): MMHal-Bench is a specialized evaluation framework designed to assess hallucination in LVLMs using 96 challenging queries derived from OpenImages, employing GPT-4 (`gpt-4-0613`) to grade responses on a scale of 0 to 5. We report both the average score and the hallucination rate (Hall.). Specifically, the hallucination rate is calculated by treating a score of 3 as the threshold for factual correctness; responses scoring below 3 are considered to contain hallucinations.

## C. Details of Visual Absence Simulation

To quantify VIG, we compare the model's loss with and without visual information. Following Xing et al. (2025), we simulate visual absence by applying a Gaussian blur with hyperparameters scaled to the input image resolution, thereby eliminating semantic visual cues. Fig. C.1a and Fig. C.1b illustrate the original image and its blurred counterpart.

## D. Details of VIG-guided Selective Training

*Table D.1.* Selection threshold $\tau_p$ at $p = 70$ across the models.

| Model | LLaVA-1.5 7B | LLaVA-1.5 13B | ShareGPT4V 7B | Open-Qwen2VL 2B |
|---|---|---|---|---|
| Threshold $\tau_{70}$ | $-0.021$ | $0.046$ | $-0.042$ | $0.012$ |

*Table D.2.* Hyperparameters for training, which are identical to the original models and training time.

| | Model | Epoch | Batch size | LR | LR schedule | LR warmup ratio | Optimizer | Training time |
|---|---|---|---|---|---|---|---|---|
| Pretraining | ShareGPT4V | 1 | 256 | 2e-5 | cosine decay | 0.03 | AdamW | 12 hours |
| Instruction Tuning | LLaVA-1.5 Family, ShareGPT4V | 1 | 128 | 2e-5 | cosine decay | 0.03 | AdamW | 7 hours |
| | Open-Qwen2VL 2B | 1 | 128 | 2e-5 | linear-warmup + cosine decay | 0.03 | AdamW | 10 hours |

In this section, we provide the implementation details of VIG-based training for Sec. 4 to ensure clarity and reproducibility.

### D.1. Instruction-tuning Datasets and the Threshold $\tau_p$

For the LLaVA-1.5 family, we employ the instruction-tuning dataset proposed by Liu et al. (2024b). In the case of ShareGPT4V, we adhere to the original protocol by substituting the 'detailed description' samples within the LLaVA dataset with the high-quality captions produced by ShareGPT4V (Chen et al., 2025a). For Open-Qwen2VL 2B, we follow the original supervised fine-tuning setup and use a 1M random subset of the MAmmoTH-VL-10M instruction-tuning dataset. The specific selection thresholds $\tau_p$ derived at $p = 70$ across the models are as shown in Tab. D.1.

*Table D.3.* **Examples of models' responses from LLaVA^W.** When applying VIG training, the model provides more visually-grounded responses for writing tasks with visual inputs than base LLaVA-1.5 and ShareGPT4V.

| User | Describe this photo in detail. |
|---|---|
| **LLaVA-1.5 7B** | **Vanilla** — The image features a painting of a dog dressed up as a woman, wearing a dress and a hat. The dog is the main focus of the painting, and it appears to be the subject of the artwork. The painting is set in a room with a dining table visible in the background. There are also two people in the scene, one on the left side and another on the right side of the painting. The person on the left is partially visible, while the person on the right is more prominently featured in the scene. |
| | **+VIG Training** — The image features a painting of a dog wearing a **Renaissance-style dress**, complete with a hood and a cape. The dog is sitting in a pose that **resembles a woman**, giving the impression of a human-like figure. The dog's attire and the way it is positioned in the painting create a unique and intriguing scene. |
| **LLaVA-1.5 13B** | **Vanilla** — The image is a painting featuring a dog dressed up as Mona Lisa, wearing a bandana on its head and holding its hands together, posing for the portrait. The background includes mountains and a river, adding to the composition's artistic appeal. In addition to the dog, there are three people present in the scene. One person is positioned near the left edge, another person can be seen at the bottom right corner, and the third person is located on the right side of the painting. |
| | **+VIG Training** — The image features a painting of a dog dressed as the famous painting of the Mona Lisa. The dog is wearing a bandana on its head and is positioned in a way that resembles the **iconic pose of the Mona Lisa**. The painting is quite detailed, capturing the essence of the original artwork while incorporating the unique twist of the dog as the subject. The dog's facial expression and posture are reminiscent of the Mona Lisa's enigmatic smile and pose, making the painting a creative and amusing take on the **classic artwork**. |
| **ShareGPT4V 7B** | **Vanilla** — The image presents a surrealistic portrait of a dog, painted in the style of the Renaissance period. The dog, the central figure of the painting, is adorned with a red hat and a blue robe, adding a touch of whimsy to the otherwise realistic style. The background is a landscape, painted in a realistic style, with a river meandering through it, adding depth to the composition. The painting is executed in oil on canvas, a medium that allows for rich colors and intricate details. The genre of the painting is surrealism, a style that combines elements of the real and the imaginary, creating a dreamlike quality. The painting is a testament to the artist's skill in blending realism with surrealism, creating a unique and captivating piece of art. |
| | **+VIG Training** — The image presents a unique piece of art, reminiscent of the famous painting **Mona Lisa by Leonardo da Vinci**. The central figure is a dog, adorned with a **black turban** and a long, flowing **brown robe**. The dog's gaze is directed straight at the viewer, creating a sense of connection. The background is a landscape, featuring a river and mountains, adding depth to the composition. The art style is realistic, with meticulous attention to detail evident in the dog's features and the surrounding environment. The genre of the artwork is portraiture, focusing on the depiction of the **dog as the main subject**. The overall composition and style suggest a fusion of classical art techniques with a modern twist. |

*Table D.4.* **Examples of models' responses from MMVet.** When applying VIG training, the model provides more accurate responses for writing tasks with visual inputs than base LLaVA-1.5 7B.

| User | | Is this airplane taking off or landing? |
|---|---|---|
| LLaVA-1.5 7B | Vanilla | Based on the image, it is not possible to definitively determine whether the airplane is taking off or landing. However, we can observe that the airplane is in the process of either ascending or descending, as it is flying low over the runway. The presence of smoke coming from the airplane's wheels suggests that it could be either taking off or landing. To determine the exact phase of the flight, we would need more information or a clearer view of the airplane's position relative to the ground. |
| | +VIG Training | The airplane is **taking off from the runway**. |
| LLaVA-1.5 13B | Vanilla | The airplane in the image is taking off from the runway. |
| | +VIG Training | The airplane in the image is taking off from the runway. |
| ShareGPT4V 7B | Vanilla | The airplane is taking off from the runway. |
| | +VIG Training | The airplane is taking off from the runway. |

## D.2. Implementation Details

**Pretraining.** We use the official pretrained checkpoints (not instruction-tuned) of the LLaVA-1.5 family and Open-Qwen2VL 2B released on Hugging Face[1]. For ShareGPT4V 7B, we pretrain the model following the training configuration proposed in Chen et al. (2025a). The detailed training configuration for ShareGPT4V 7B is provided in Tab. D.2. Using these pretrained models, we compute the VIG score for each training sample and select samples to be used for subsequent instruction-tuning.

**VIG Calculation.** We utilize the pretrained model to compute VIG scores of the training samples. On a setup with 8 RTX 4090 (24GB) GPUs, this process takes approximately 6 hours without specific inference optimizations (e.g., vLLM).

**VIG-guided Selective Training.** Based on the data selected via VIG, we perform instruction tuning with VIG training. During this stage, we adopt the same hyperparameter settings as those used in the original implementations of LLaVA-1.5 and ShareGPT4V. We use 8 A100 (80GB) GPUs. Training details for VIG training are summarized in Tab. D.2.

## D.3. Qualitative Results

We provide qualitative results of VIG-training. We examine how VIG-guided selective training mitigates hallucinations and enhances visual grounding across different models, LLaVA-1.5 7B, 13B, and ShareGPT4V 7B.

As shown in Tab. D.3, both the vanilla LLaVA-1.5 7B and 13B models suffer from severe object hallucinations. The 7B model fabricates 'a dining table' and 'two people', while the 13B model hallucinates 'three people' in the background. These errors suggest that the models are retrieving generic descriptions associated with classical paintings or indoor scenes from their trained knowledge rather than referring to the specific input image. In contrast, the VIG-trained models successfully suppress these text-driven fabrications. By filtering out low-VIG tokens during training, our method encourages the model to

---

[1]LLaVA-1.5 7B: https://huggingface.co/liuhaotian/llava-v1.5-mlp2x-336px-pretrain-vicuna-7b-v1.5
LLaVA-1.5 13B: https://huggingface.co/liuhaotian/llava-v1.5-mlp2x-336px-pretrain-vicuna-13b-v1.5
Open-Qwen2VL 2B: https://huggingface.co/weizhiwang/Open-Qwen2VL-base

verify visual existence before generation, resulting in faithful descriptions that accurately capture the dog's solitary presence and specific pose.

Beyond object existence, VIG training significantly improves the precision of attribute recognition. The vanilla ShareGPT4V 7B model, despite being a stronger baseline than LLaVA-1.5 7B, is susceptible to linguistic shortcuts. Recognizing the visual similarity to the Mona Lisa, the model defaults to describing the original painting's attributes—hallucinating a 'red hat' and 'blue robe'. This indicates a blind faith in the semantic concept over pixel-level evidence. The VIG-trained model effectively breaks this shortcut. By prioritizing tokens with high visual information gain, the model correctly grounds the attributes in the actual image, accurately identifying the 'black turban' and 'brown robe'. This demonstrates that VIG-guided training forces the model to override misleading textual priors with genuine visual evidence, leading to more robust and grounded multimodal generation.

Furthermore, Tab. D.4 presents an illustrative example from the MMVet, suggesting that VIG training can help smaller models achieve response quality comparable to larger or stronger baselines in certain scenarios. In this case, when asked to determine the airplane's action, the vanilla LLaVA-1.5 7B model exhibits uncertainty, providing a lengthy and ambiguous description. In contrast, the VIG-trained model delivers a concise and accurate answer ("taking off"), which aligns with the outputs of the significantly larger LLaVA-1.5 13B and the more advanced ShareGPT4V 7B. This observation implies that VIG training has the potential to enhance the visual grounding capability of smaller models.

## E. Additional Analysis

In this section, we extend the analysis presented in Sec. 4.4 to LLaVA-1.5 13B and ShareGPT4v 7B.

### E.1. Visual Attention Ratio

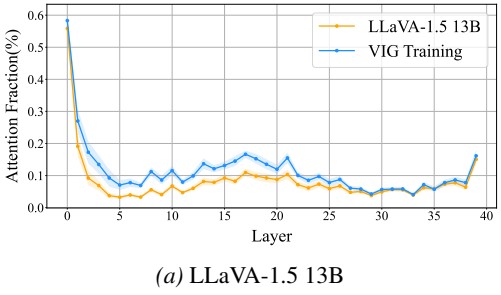 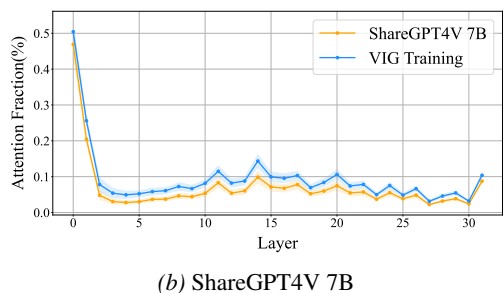

*(a)* LLaVA-1.5 13B  *(b)* ShareGPT4V 7B

*Figure E.1.* **Attention fraction allocated to visual tokens.** Compared to their respective baselines, VIG-trained models consistently assign significantly higher attention weights to visual tokens across all layers, demonstrating improved visual grounding regardless of model scale or architecture.

Fig. E.1a and Fig. E.1b illustrate the proportion of attention weights allocated to visual tokens across all layers for each model. Regardless of the model scale or the baseline architecture, a common trend is observed: the VIG-trained models consistently assign significantly higher attention scores to visual tokens compared to their original counterparts.

### E.2. Blind Faith in Text

Fig. E.2a presents the results for LLaVA-1.5 13B. While the clean accuracy (Base) remains comparable, the VIG-trained model shows significantly higher robustness under corruption (42.9%) compared to the vanilla model (36.8%), resulting in a notable improvement in the normalized score (Norm) from 45.7 to 53.0. A similar trend is observed with ShareGPT4V 7B in Fig. E.2b. The vanilla model suffers a severe performance drop when facing corrupted text (30.5%). In contrast, VIG training boosts the corruption accuracy to 39.3% and improves the normalized score by over 10%. These results consistently demonstrate that VIG-guided training effectively mitigates language bias and encourages the model to ground its predictions in visual evidence.

To further illustrate that VIG improves visual grounding fundamentally at the representational level, we compare it with VCD under the same text corruption setting using LLaVA-1.5 7B. As shown in Tab. E.1, VCD yields only a marginal improvement under text corruption (+1.7 in Norm score). In contrast, VIG training produces a substantially larger gain (+12.9 in Norm score). This confirms that explicitly addressing language bias within the training data is a more fundamental

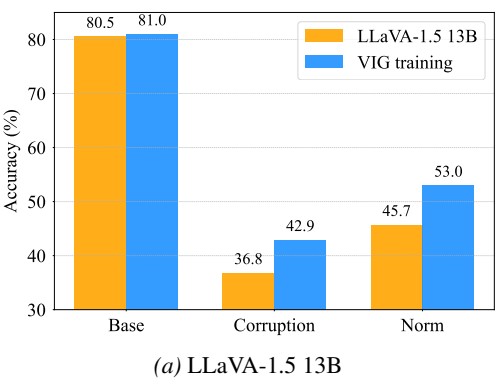
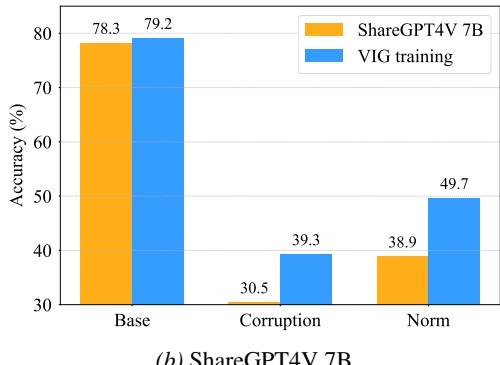

*(a)* LLaVA-1.5 13B          *(b)* ShareGPT4V 7B

*Figure E.2.* **Evaluation of text reliance under textual corruption.** Base: accuracy on clean inputs. Corruption: accuracy when the same image is paired with a corrupted caption containing a conflicting description. Norm: corruption accuracy normalized by the corresponding Base (Corruption/Base).

*Table E.1.* **Comparison with VCD under textual corruption on VQAv2.** Evaluated on LLaVA-1.5 7B. "Base" represents accuracy on clean inputs, while "Corruption" is the accuracy when misleading text is present. "Norm" is the normalized score (Corruption / Base × 100).

| Model | Base | Corruption | Norm |
|---|---|---|---|
| LLaVA-1.5 7B (Baseline) | 77.9 | 32.1 | 41.2 |
| + VCD | 78.1 | 33.5 | 42.9 |
| + VIG training | **78.2** | **42.3** | **54.1** |

solution than inference-time logit penalties, as it enables the model to genuinely learn to resist spurious textual cues.

### E.3. Text comprehension

*Table E.2.* **Comparison of text-only benchmark performance before and after VIG-guided training.** For both LLaVA-1.5 13B and ShareGPT4V, performance on GSM8K, MMLU, HellaSwag, and TruthfulQA remains largely unchanged after VIG training, suggesting that improvements in visual grounding do not come at the expense of text comprehension.

| Model | GSM8K | MMLU | HellaSwag | TruthfulQA |
|---|---|---|---|---|
| LLaVA 1.5 13B | 15.33 | 51.68 | 76.09 | 45.86 |
| + VIG training | 14.99 | 51.35 | 76.11 | 46.01 |
| ShareGPT4V 7B | 18.01 | 47.62 | 54.78 | 46.53 |
| + VIG training | 18.03 | 47.66 | 56.14 | 46.42 |

We present text-only benchmark results in Tab. E.2 to assess whether VIG-guided training affects general language understanding. Across GSM8K, MMLU, HellaSwag, and TruthfulQA, both LLaVA-1.5 13B and ShareGPT4V maintain comparable performance before and after VIG training. The observed differences are small overall, indicating that VIG-guided training preserves text comprehension while improving visual grounding.

### E.4. Impact of VIG-based Filtering on Data Distribution

To understand how VIG-guided selective training alters the composition of the instruction-tuning dataset, we conduct a distribution analysis on the selected 70% of the LLaVA-665K dataset. We investigate whether the filtering strategy disproportionately removes rare concepts. Specifically, we assess the long-tail distribution of COCO object categories in the selected data. The object categories are grouped into three frequency tiers: head (top 20%), torso (middle 60%), and tail (bottom 20%). As shown in Fig. E.3a, the retention rates remain remarkably uniform across all tiers—63.0% for Head, 65.8% for Torso, and 66.1% for Tail, which indicates that rare concepts are well-preserved. While the conceptual coverage remains stable, we observe a significant shift at the task level in the dataset. As shown in Fig. E.3b, tasks that strictly require dense visual grounding, such as OCR-VQA and TextVQA, exhibit extremely high retention rates. Conversely, tasks where

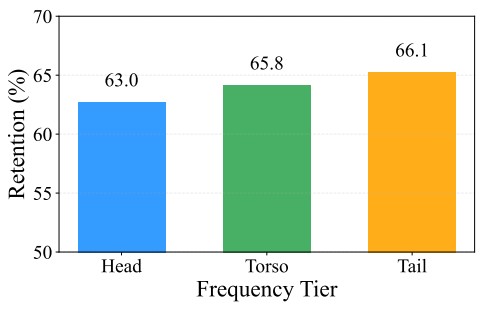
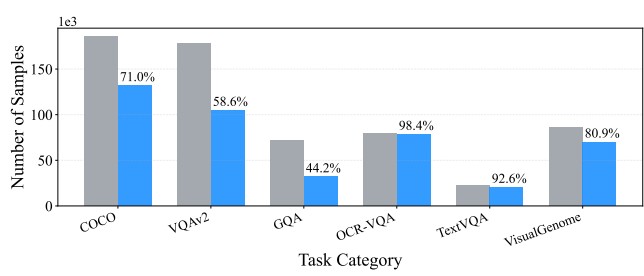

*(a)* Long-tail category coverage     *(b)* Task-level distribution shift

*Figure E.3.* **Data Distribution after VIG-based Filtering.** Compared to the original data distribution, VIG-based filtering consistently preserves long-tail samples while selectively reshaping the task composition of the training data. This suggests that VIG-based filtering improves sample quality without sacrificing broad data coverage.

models can often guess the answer using textual priors or common sense (e.g., VQAv2, GQA) are filtered much more aggressively. These findings demonstrate that VIG-based selection evaluates visual necessity at the sample level rather than simply filtering out rare words, ensuring that long-tail concepts and conceptual diversity are preserved in the training data.

## F. Details of Ablation Study

### F.1. Effectiveness of VIG-based Selection

#### F.1.1. COMPREHENSIVE RESULTS

Tab. F.1 provides comprehensive results for the experimental results of the selection level. Notably, *Random* degrades performance across most benchmarks compared to the baseline, confirming that simply reducing data volume is detrimental. In contrast, the combination of sample and token-level selection (*SS+TS*) achieves the best results, significantly outperforming the baseline.

*Table F.1.* **Ablation study of selection levels on LLaVA-1.5 7B.** "Random" trains on a random 70% subset of the data, "SS" selects the top 70% samples by VIG score (sample-level selection only), and "SS+TS" additionally applies token-level VIG selection. For each metric, ↑ indicates higher is better and ↓ indicates lower is better. **Bold** indicates the best performance.

| Model | LLaVA$^W$ | Vision Understanding MMVet | MMBench | DocVQA | POPE F1 ↑ | Acc. ↑ | Hallucination CHAIR $C_S$ ↓ | $C_I$ ↓ | MMHal Score ↑ | Hall. ↓ |
|---|---|---|---|---|---|---|---|---|---|---|
| | | Score ↑ | | Acc. ↑ | | | | | | |
| LLaVA-1.5 7B | 59.02 | 28.62 | 65.46 | 22.31 | 85.90 | 87.08 | 52.93 | 14.99 | 1.71 | 71.25 |
| Random | 58.50 | 27.27 | 55.97 | 21.38 | 85.48 | 86.63 | 53.11 | 15.00 | 1.79 | 71.33 |
| SS | 58.71 | 32.29 | 57.56 | 22.06 | 85.22 | 86.53 | 52.10 | 14.61 | 1.84 | 63.54 |
| SS+TS | **61.22** | **32.71** | **66.33** | **22.51** | **85.93** | **87.47** | **47.00** | **12.80** | **2.23** | **62.78** |

### F.2. Effect of Selection Ratio

#### F.2.1. THE THRESHOLD $\tau_p$ VALUES

To investigate the impact of the selection ratio, we conduct experiments with $p \in \{30, 50, 70, 100\}$ on LLaVA-1.5 7B. Note that $p = 100$ corresponds to the vanilla model trained on the full instruction-tuning dataset without VIG-based selection. Selection thresholds $\tau_p$ derived at each $p$ are as shown in Tab. F.2.

#### F.2.2. COMPREHENSIVE RESULTS

Tab. F.3 demonstrates that VIG-guided filtering ($p < 100$) consistently outperforms the full-data baseline on hallucination and open-ended benchmarks. While the aggressive setting ($p = 30$) offers extreme efficiency (using only ∼5% of tokens), it shows minor degradation on broad-coverage tasks like MMBench. In contrast, the $p = 70$ configuration achieves the

*Table F.2.* Selection threshold $\tau_p$ at each $p$ on LLaVA-1.5 7B.

| Ratio $p$ | 30 | 50 | 70 |
|---|---|---|---|
| Threshold $\tau_p$ | 0.124 | 0.031 | $-0.021$ |

optimal balance, securing peak scores on POPE and MMHal by preserving sufficient diversity for complex reasoning while still significantly reducing computational cost.

*Table F.3.* Ablation study of selection ratio $p\%$ on LLaVA-1.5 7B. "# Sample Tokens" represents the total number of answer tokens contained in the multimodal samples retained after sample-level selection. "# Active Tokens" refers to the effective number of tokens that contribute to the loss computation after applying token-level masking. For each metric, ↑ indicates higher is better and ↓ indicates lower is better. **Bold** indicates the best performance.

| Ratio $p$ | # Sample Tokens | # Active Tokens | LLaVA$^W$ | MMVet | MMBench | DocVQA | POPE | | CHAIR | | MMHal | |
|---|---|---|---|---|---|---|---|---|---|---|---|---|
| | | | | Score ↑ | | Acc. ↑ | F1 ↑ | Acc. ↑ | $C_S$ ↓ | $C_I$ ↓ | Score ↑ | Hall. ↓ |
| 30% | 8.58M | 3.19M | 60.90 | **34.03** | 56.87 | **22.66** | 82.80 | 84.28 | 48.71 | 13.31 | 1.99 | 63.01 |
| 50% | 26.32M | 10.00M | **61.81** | 31.19 | 56.44 | 22.49 | 85.09 | 86.14 | 47.63 | 12.99 | 1.95 | 62.91 |
| 70% | 51.17M | 38.45M | 61.22 | 32.71 | **66.33** | 22.51 | **85.93** | **87.47** | **47.00** | **12.80** | **2.23** | **62.78** |
| 100% | 58.61M | 58.61M | 59.02 | 28.62 | 65.46 | 22.31 | 85.90 | 87.08 | 52.93 | 14.99 | 1.71 | 71.25 |

