# OpenReview forum: "Focusing Where Vision Matters: Selective Training for Large Vision Language Models via Visual Information Gain"
_ICML.cc/2026/Conference — ICML 2026 regular_

### Official Review · Reviewer_LQzP · 2026-03-02

**Soundness:** 3
**Presentation:** 3
**Significance:** 3
**Originality:** 3
**Overall Recommendation:** 4
**Confidence:** 4

**Summary:**

This paper focuses on the language bias problem of LVLMs. They proposes VIG, a metric which effectively measures the visual gain of the sample by calculating the loss difference with and without the input image. Furthermore, the paper extends VIG to tokens, and select the samples and tokens with the highest VIG scores for training. Compared with vanilla models, the trained models are improved across various visual understanding tasks.

**Compliance With Llm Reviewing Policy:**

Affirmed.

**Final Justification:**

I raise my score to weak accept. The paper proposes VIG, a simple and effective metric for quantifying visual importance in multimodal training data, with clear motivation and strong empirical support. My main concerns were about generalizability to newer architectures, impact on text comprehension, and what characterizes visually important data. The rebuttal addressed all three convincingly: the Open-Qwen2VL experiment demonstrates generality beyond LLaVA, text-only benchmarks confirm no degradation, and the data construction guidelines offer useful insights for the community. These additions meaningfully strengthen the paper.

**Key Questions For Authors:**

1. The LLaVA-1.5 model and the ShareGPT4V-7B model were both released a few years ago, and their architectures are also LLaVA architecture. Is it possible to add new LVLMs for training and evaluation? For example, QwenVL series, InternVL series. Otherwise, the current work seems to only show that the quality of the instruction tuning data of these two models are not high, and may not necessarily solve the current problem of new LVLMs.
2. After training with only more visually important data, how does the model perform on data that does not have such a large visual information gain? What about text-only tasks? Is this training effect achieved at the expense of text comprehension?
3. What is the difference between visually important data and unimportant data? Does this difference come from the image itself, or from the difficulty of the question? Can you give the community some inspiration on selecting images and structuring problems?
If the questions are properly answered, I will consider raising my score.

**Limitations:**

No.
The model used in the article is relatively early, and the data sets are all visual understanding tasks. It may have limited effect on newer models and text-only tasks.

**Strengths And Weaknesses:**

- **Soundness**: The motivation and logic of the article are very clear. The design of VIG is simple and effective, and is supplemented by a large number of ablation and analysis experiments to prove the effectiveness of the method.
- **Presentation**: The writing of the article is very clear and easy to read.
- **Significance**: Language bias is a common problem, and the VIG method proposed in the paper identifies which samples are more visually important from the data perspective, and obtains better results through efficient training methods, which has highly practical value.
- **Originality**: It is very novel to analyze the language bias phenomenon from the perspective of data and be the first to propose the visually important metric. Combining data analysis with selective training is an interesting attempt.

---

> ### Author Rebuttal · Authors · 2026-03-30
>
> Thank you for your thoughtful review and suggestions. We address each point below and will include the additional experiments and discussions in the revised manuscript.
>
> **Q1: Generalizability to modern LVLM architectures**
>
> VIG requires full reproducibility of the training pipeline, specifically, access to both the alignment-only pretrained checkpoint and the instruction tuning dataset. InternVL's SFT data has been withdrawn due to company policy (see huggingface.co/datasets/OpenGVLab/InternVL-Data), and Qwen-VL's training data remains proprietary. Under these constraints, LLaVA-1.5 and ShareGPT4V represent the most suitable choices, as their training pipelines are fully transparent and reproducible.
>
> To directly address the generalizability concern, we conducted additional experiments on Open-Qwen2VL (2B) [R1], which adopts a substantially different architecture from LLaVA: SigLIP encoder, Qwen2.5 LLM, and Adaptive Average-Pooling projector. Its pretrained checkpoint, SFT data, and training code are all publicly available. Given the limited rebuttal period, we sampled 1M data points from the MAmmoTH-VL 10M dataset and applied VIG training ($p=70$).
>
> ||Sample Tokens|Active Tokens|LLaVAW|MMVet|MMBench|DocVQA|POPE (F1)|POPE (Acc)|CHAIR (C_S)|CHAIR (C_I)|MMHal (Score)|MMHal (Hal.)|
> |---|---|---|---|---|---|---|---|---|---|---|---|---|
> |Open-Qwen2VL|4.10B|4.10B|63.89|37.77|78.59|41.05|86.48|87.27|29.84|7.55|1.99|64.28|
> |+ VIG training|**3.29B**|**2.42B**|**65.17**|**39.01**|**79.67**|**44.05**|**87.99**|**87.98**|**27.74**|**6.98**|**2.23**|**62.56**|
>
>
> As shown in the table above (**Bold** indicates the better performance), despite using 41% fewer active tokens, VIG training improves performance across all benchmarks. This confirms that VIG generalizes beyond LLaVA-based architectures. It also suggests that recently proposed training datasets still contain sources of language bias. In other words, the issue is not specific to LLaVA's data quality, and VIG-based filtering remains applicable to newer architectures and datasets as well.
>
> [R1] Wang et al., Open-Qwen2VL: Compute-Efficient Pre-Training of Fully-Open Multimodal LLMs on Academic Resources, COLM 2025
>
> **Q2: Impact on text comprehension**
>
> VIG is designed to avoid degrading text capabilities through two mechanisms. First, VIG-based selection is applied exclusively to the multimodal subset of the dataset: text-only instruction tuning samples are kept entirely intact. Second, during token-level selection, unselected tokens are masked from the loss computation rather than removed from the input sequence, so the full context is preserved during training.
>
> ||GSM8K|MMLU|HellaSwag|TruthfulQA|
> |---|---|---|---|---|
> |LLaVA 1.5 7B|15.33|51.68|76.09|45.86|
> |+ VIG training|14.99|51.35|76.11|46.01|
> |LLaVA 1.5 13B|26.69|56.89|80.36|43.35|
> |+ VIG training|26.00|57.19|79.78|43.54|
>
> To directly verify this, we evaluated VIG-trained models on standard text-only LLM benchmarks. As shown above, performance is well maintained across all metrics, confirming that VIG's gains in visual grounding are not achieved at the expense of text comprehension.
>
> **Q3: What makes data visually important and implications for data construction**
>
> VIG is determined not by image complexity or question difficulty alone, but by their interaction: how much the image reduces prediction uncertainty for a given QA pair. Tab. 1 demonstrates this: fixing the QA pair and varying the image changes VIG from 0.923 (matched) to −0.520 (conflicting). The reverse also holds: for the same image, "describe this photo" (requiring visual details) yields high VIG, while "what is the trail behind a boat called?" (answerable from common sense) yields low VIG (Fig. 1a vs. 1b). We will include additional examples in the revised appendix.
>
> From this analysis, we can offer concrete guidelines for multimodal data construction:
>
> - Question design: Questions targeting visually grounded attributes (e.g., color, spatial relations, object states) produce higher VIG and contribute more effectively to visual grounding during training (Tab. 2). Questions answerable from common sense alone contribute less.
> - Image-question coupling: Even with a visually rich image, a question that does not require the image will have low VIG. Effective data curation should ensure the question genuinely depends on the visual content.
> - VIG as a data filter: VIG can be applied post-hoc on any candidate instruction dataset to quantify visual dependency before training, helping practitioners identify weakly grounded samples without manual inspection.

---

> > ### Author Rebuttal · Reviewer_LQzP · 2026-04-01
> >
> > I will raise my score.

---

> > > ### Author Response · Authors · 2026-04-05
> > >
> > > We sincerely thank the reviewer for the thoughtful questions throughout the discussion. The additional experiments on Open-Qwen2VL, text-only benchmarks, and the data construction guidelines have substantially improved the paper. We will incorporate all additions into the revised manuscript.

---

### Official Review · Reviewer_qcKu · 2026-03-04

**Soundness:** 2
**Presentation:** 3
**Significance:** 2
**Originality:** 1
**Overall Recommendation:** 2
**Confidence:** 5

**Summary:**

The authors focus on an important problem that LVLMs equally treat vision and context tokens, and thus leading to visual ignorance and hallucinations. Then, the authors propose a metric named Visual Information Gain, aiming to measure how much the instruction data sample relies on the visual information. This is an interesting research direction, but several limitations still exist as in the parts below.

**Compliance With Llm Reviewing Policy:**

Affirmed.

**Final Justification:**

Please refer to ack.

**Key Questions For Authors:**

Please see weaknesses.

**Limitations:**

N/A.

**Strengths And Weaknesses:**

Strength:

1-The research motivation is interesting and important. Different training samples show different characteristics, and some of them mainly rely on the vision information. But existing LVLMs treat all samples equally and may ignore the vision leading to hallucinations.

2-The writing is clear and easy to follow.

Weakness:

1-The proposed VIG is too simple. It utilizes a direct difference between (A|Q) and (A|Q, I), which is similar to previous works, such as VCD (Leng 2024) and ALFAR [R1].

[R1] Boosting Knowledge Utilization in Multimodal Large Language Models via Adaptive Logits Fusion and Attention Reallocation. NeurIPS 2025.

2-The significance of VIG is not strong. The analysis is straightforward and aligns with common sense. The VIG, alongside its analysis, does not bring many insights and new findings.

3-The experiments are not enough. Only 2 models (LLaVA and ShareGPT) are tested. There are many new LVLMs, but the authors only choose these outdated ones.

4-The performance gains are limited. It does not surpass the training-free methods by a clear margin with costly fine-tuning.

5-I recommend the authors to design some training-free manners based on the VIG metric.

---

> ### Author Rebuttal · Authors · 2026-03-30
>
> We appreciate the detailed feedback. We address each point below and will include the additional experiments and discussions in the revised manuscript.
>
> **W1: Novelty of VIG relative to VCD and ALFAR**
>
> We first note that ALFAR is conceptually distinct from both VIG and VCD. ALFAR does not compare predictions with and without visual input: it compares logits with and without RAG-retrieved contextual knowledge to maximize the utilization of retrieved text. The problem domain (retrieval-augmented generation) is fundamentally different from visual grounding in standard VLM training.
>
> VIG does share with VCD the use of a probability gap between visual and non-visual conditions. The distinction lies in how this signal is applied. VCD uses it at inference time to correct model outputs, leaving the model itself unchanged. VIG uses it at training time to select which samples and tokens receive gradients, thereby improving the model's fundamental visual grounding ability. This is why the two methods are complementary. Tab. 4 shows consistent additive gains when VCD is applied on top of a VIG-trained model. We refer the reviewer to our response to Reviewer `g14S` (`W1`) for further discussion on this point.
>
> **W2: Practical significance of VIG**
>
> VIG's contribution is not to confirm that certain tokens are visually dependent. It is to quantify this dependency at scale and make it actionable for training. Without such quantification, selective sample- and token-level training (Sec. 3.4) would not be possible. The practical value lies in enabling automatic filtering of large-scale supervision without manual labeling.
>
> Our analysis also reveals findings that are not self-evident. For instance, GQA and ScienceQA (widely treated as visual benchmarks) exhibit substantial text dependence with negative mean VIG (Fig. 2). This offers a critical perspective on current benchmark design that goes beyond common intuition. We also conducted a corpus-level POS analysis confirming systematic patterns: proper nouns show the highest mean VIG (0.514), followed by nouns (0.144) and verbs (0.106), while function tokens cluster near zero. We refer the reviewer to our response to Reviewer `g14S` (`Q1`) for details.
>
> **W3: Generalizability of VIG training**
>
> As shown in our response to Reviewer `LQzP` (`Q1`), we applied VIG training to Open-Qwen2VL (2B) [R1], a recent LVLM with a fundamentally different architecture from LLaVA: SigLIP vision encoder, Qwen2.5 LLM backbone, and Adaptive Average-Pooling projector. We chose this model because it is one of the few modern LVLMs whose pretrained checkpoint, SFT data, and training code are all publicly available: a prerequisite for applying VIG, which requires full reproducibility of the training pipeline. For reference, InternVL's SFT data has been withdrawn due to company policy, and Qwen-VL's training data remains proprietary.
>
> VIG training ($p=70$) on Open-Qwen2VL consistently improves all benchmarks despite using 41% fewer active tokens. This confirms that VIG generalizes beyond LLaVA-based architectures. It also suggests that language bias in instruction tuning data is not specific to LLaVA's data quality: even recently curated datasets benefit from VIG-based filtering.
>
> [R1] Wang et al., Open-Qwen2VL: Compute-Efficient Pre-Training of Fully-Open Multimodal LLMs on Academic Resources, COLM 2025
>
> **W4: Performance gains relative to training-free methods**
>
> We respectfully note that the claim of "limited performance improvement" is not supported by the full results in Tab. 3. VIG training consistently outperforms the vanilla baseline across all benchmarks, with notable gains: $\mathrm{MMVet}$ +4.09, $\mathrm{CHAIR}$ $C_S$ −5.93, $\mathrm{MMHal~Hall.}$ −8.47. In contrast, training-free methods exhibit clear trade-offs: VCD drops on $\mathrm{MMVet}$ (28.62 → 27.01), LACING decreases $\mathrm{DocVQA}$ (22.31 → 21.45). VIG shows no such trade-off.
>
> Furthermore, VIG does not require costly additional fine-tuning. It selects data within the existing instruction tuning pipeline. After training, inference proceeds with zero additional overhead, whereas training-free methods like VCD require an extra forward pass per query at deployment. The two approaches are also complementary: Tab. 4 shows that combining VIG with VCD, PAI, or VAR yields additive gains, confirming they address different factors.
>
> **W5: Training-free application of VIG**
>
> We thank the reviewer for this suggestion. Our current contribution focuses on using the visual-presence signal at training time for data and token selection, which improves the model's inherent visual grounding: something inference-time methods cannot achieve. Exploring training-free applications of VIG (e.g., as an inference-time confidence signal) is an interesting future direction that we will discuss in the revised manuscript.

---

> > ### Author Rebuttal · Reviewer_qcKu · 2026-04-03
> >
> > Please refer to comments.

---

> > > ### Author Response · Authors · 2026-04-03
> > >
> > > We sincerely appreciate your time and feedback. To ensure that we address your concerns appropriately, could you indicate the specific comments that remain unresolved and the parts of our rebuttal that were insufficient? Your guidance would greatly help us provide a more precise and meaningful response.

---

### Official Review · Reviewer_zDFx · 2026-03-06

**Soundness:** 4
**Presentation:** 4
**Significance:** 3
**Originality:** 3
**Overall Recommendation:** 5
**Confidence:** 4

**Summary:**

This paper introduces VIG: a metric that assesses how much visual information is shared between the text and image of a multimodal training sample. VIG is computed by taking the difference in cross-entropy loss between the sample with and without the image. VIG can be used as an interpretability measure, where the authors show that it is informative to showing the visual dependency across different benchmarks, and also as a mechanism for improving training, where it can improve performance when used to filter data and weigh the loss.

**Compliance With Llm Reviewing Policy:**

Affirmed.

**Final Justification:**

My score was positive already and the authors provided the requested analysis on CV-bench and clarified some of my other comments.

**Key Questions For Authors:**

No questions

**Limitations:**

Yes

**Strengths And Weaknesses:**

Strengths: The method is very simple and effective. The topic area is very important since visual grounding is known to be a problem for MLLMs. I like this VIG can both be used for interpretability and improving training. The paper is well written and easy to follow. The experiments are extensive and there are both qualitative and qualitative results.

Weaknesses: One thing that I think would be valuable to test is computing VIG on CV-Bench [1] as done in Figure 2. CV-Bench is specifically designed to require the image to answer, and it so I would expect that if VIG is informative we should see that CV-Bench has a higher value. This would also be a good benchmark to evaluate the model on after training with VIG. Another experiment that I think is needed is ablating the two steps of VIG training. First, data filtering at the sample level occurs and then a loss weighting strategy is used. I would like to see an ablation of just doing data filtering and no loss weighting. I am a bit skeptical the loss weighting matters that much as many works have established the effectiveness of simple data filtering. I also think that there should be a larger gallery with many qualitative samples like in Table 1, perhaps in the appendix.


[1] Cambrian-1: A Fully Open, Vision-CentricExploration of Multimodal LLMs

---

> ### Author Rebuttal · Authors · 2026-03-30
>
> Thank you for the positive assessment and constructive suggestions. We address each point below.
>
> **W1: Analysis and evaluation on CV-Bench**
>
> We computed the VIG distribution on CV-Bench using the same methodology as Fig. 2:
>
> ||COCO|POPE|GQA|SQA|CV-Bench|
> |---|---|---|---|---|---|
> |Mean|0.29|0.01|-0.04|-0.09|0.07|
> |Var.|0.02|0.02|0.04|0.06|0.03|
>
> CV-Bench shows a positive mean VIG, reflecting higher visual reliance than GQA and SQA. This is consistent with its design as a vision-centric benchmark and further validates that VIG captures benchmark-level visual dependency.
>
> We also evaluated VIG-trained models on CV-Bench:
>
> ||2D|3D|
> |---|---|---|
> |LLaVA 1.5 7B|58.08%|60.28%|
> |+ VIG training|59.79% (+1.71)|65.16% (+4.88)|
> |LLaVA 1.5 13B|59.98%|60.34%|
> |+ VIG training|60.11% (+0.13)|65.67% (+5.33)|
>
> VIG training yields consistent gains, with particularly large improvements on 3D tasks (+4.88 / +5.33), which require stronger spatial reasoning from visual input. We will add these results to Fig. 2 and Tab. 3 in the revised manuscript.
>
> **W2: Ablation study on selection level**
>
> The requested analysis is provided in Tab. 5 and Tab. E.1 in the manuscript. SS (sample-level selection only) and SS+TS (sample + token-level selection) directly isolate the contribution of token-level masking. SS improves hallucination metrics over random selection, but does not consistently outperform the full-data baseline on vision understanding benchmarks (e.g., $\text{LLaVA}^{\text{W}}$, $\text{MMBench}$). SS+TS achieves the best results across all metrics. This gap reflects the fact that even within high-VIG samples, many tokens (articles, prepositions, etc.) carry little visual information. Token-level masking prevents these from diluting the learning signal.
>
> **W3: Additional qualitative samples**
>
> Due to the text-only format of the rebuttal, we cannot include the full gallery here, but our additional samples show the same trend as Tab. 1: higher-VIG samples exhibit strong alignment between visual content and the ground-truth answer, while low-VIG samples tend to be answerable from text alone. We will include additional qualitative examples covering diverse visual grounding types (color, spatial relations, object attributes, actions) across varying VIG levels in the revised appendix.

---

> > ### Author Rebuttal · Reviewer_zDFx · 2026-04-02
> >
> > My score was positive already and the authors provided the requested analysis on CV-bench and clarified some of my other comments.

---

> > > ### Author Response · Authors · 2026-04-05
> > >
> > > We sincerely thank the reviewer for the constructive suggestions throughout the discussion. The additional experiments on CV-Bench and the expanded qualitative analysis have meaningfully strengthened the paper. We will incorporate all additions into the revised manuscript.

---

### Official Review · Reviewer_g14S · 2026-03-12

**Soundness:** 3
**Presentation:** 3
**Significance:** 3
**Originality:** 2
**Overall Recommendation:** 4
**Confidence:** 4

**Summary:**

To address the challenge of text-dominance in multimodal training data, this paper introduces VIG, a perplexity-based metric designed to assess the importance of visual input. By leveraging VIG to curate high-quality vision-language data, the authors demonstrate its effectiveness in improving visual grounding, mitigating language bias across various understanding tasks, and reducing model hallucinations.

**Compliance With Llm Reviewing Policy:**

Affirmed.

**Final Justification:**

The manuscript underscores the significance of addressing language bias during the data selection phase. Nevertheless, since the methodology and analysis rely heavily on established frameworks, I suggest a Weak Accept.

**Key Questions For Authors:**

* Does the loss difference ($y - x$) in Figure 3 exhibit any consistent patterns across specific token types?

**Limitations:**

The potential impact of this filtering method on the data distribution should be discussed.

**Strengths And Weaknesses:**

### Strengths
* The proposed method is straightforward yet effective.
* The manuscript is well-organized and easy to follow.
* The work has the potential to serve as a benchmark/guideline for constructing high-quality vision-language alignment datasets.


### Weaknesses
* Comparing standard and impaired visual inputs is a common practice for quantifying linguistic bias and visual reliance (e.g., VCD). The paper needs to further clarify its technical departure from existing methods to justify its novelty.
* It remains unclear how low-quality VIG training data contributes to model hallucinations. Does the model simply memorize the incorrect answers, or does it develop flawed text-visual alignments during the training process?
* In contrast to training-free approaches such as VCD, the proposed method necessitates an additional fine-tuning stage, which introduces significant computational overhead and resource requirements
* The paper lacks a discussion on whether VIG-based filtering leads to category imbalance or shifts in the long-tail distribution of the dataset.

---

> ### Author Rebuttal · Authors · 2026-03-30
>
> Thank you for the insightful and constructive feedback. We address each point below and will include the additional analyses in the revised manuscript.
>
> **W1: Novelty beyond standard vs. impaired input comparison**
>
> Comparing model outputs with standard and impaired visual inputs is a common technique in the VLM literature, including VCD. What distinguishes VIG is how this signal is used. VCD applies it at inference time to correct model outputs, but the model itself remains unchanged. VIG instead uses it for training-time data and token selection (Eqs. 2–6), improving the model's visual grounding ability through selective training. This is why the two methods are complementary. Tab. 4 shows consistent additive gains when VCD is applied on top of a VIG-trained model, which would be unlikely if they addressed the same factor.
>
> **W2: How low-VIG data contributes to hallucination**
>
> We argue that the issue is not memorization of wrong answers but a gradual weakening of visual-textual alignment. In low-VIG samples, the correct answer can be predicted from text alone, so the model is rewarded for linguistic shortcuts rather than visual attention. Supporting evidence: (1) many tokens such as articles and prepositions carry near-zero VIG, and training them with equal loss weight lets text-driven patterns dominate (Tab. 2); (2) VIG-trained models allocate substantially more attention to visual tokens (Fig. 4); (3) VIG-trained models better resist misleading textual cues (Fig. 5). Together, these suggest that low-VIG data contributes to hallucination by reinforcing linguistic shortcuts that produce plausible but visually ungrounded content.
>
> **W3: Computational overhead vs. training-free methods**
>
> We clarify that VIG does not introduce an additional fine-tuning stage. The instruction tuning procedure itself remains unchanged. VIG only adds a data selection step beforehand. The additional cost is one-time VIG scoring (~6 hours on 8 RTX 4090s for 7B models, fully parallelizable), after which instruction tuning proceeds as usual but on a smaller subset (30% less). Active tokens drop by 34-79% depending on the model (Tab. 3), and the additional inference cost is zero. Training-free methods like VCD require an additional forward pass per query at every inference. Note again that VIG is complementary rather than competitive with training-free methods, as shown in Tab. 4.
>
> **W4: Impact of VIG-based filtering on data distribution**
>
> We analyzed the LLaVA 665k instruction-tuning dataset after filtering at $p=70$ along two axes.
> - Long-tail category coverage: We examine COCO object category retention rates by frequency tier (Head: top 20%, Torso: middle 60%, Tail: bottom 20%). As shown below, retention is nearly uniform across tiers, confirming that VIG-based filtering does not disproportionately remove tail categories.
> 	|                | Head  | Torso  | Tail  |
> 	| -------------- | ----- | ------ | ----- |
> 	| Retention rate | 63.0% | 65.8 % | 66.1% |
>
> - Task-level distribution shift: Retention rates vary across task types. As shown below, tasks requiring dense visual grounding (OCR-VQA, TextVQA) show high retention, while those with weaker visual dependency (VQAv2, GQA) are filtered more aggressively.
> 	|                | OCR-VQA | TextVQA | VisualGenome | COCO Caption | VQAv2 | GQA   |
> 	| -------------- | ------- | ------- | ------------ | ------------ | ----- | ----- |
> 	| Retention rate | 98.4%   | 92.6%   | 80.9%        | 71.0%        | 58.6% | 44.2% |
>
> These results confirm that VIG shifts the data mix toward visually grounded samples based on actual visual necessity, without compromising long-tail coverage.
>
> **Q1: Consistent patterns in token-level loss difference (Fig. 3)**
>
> We conducted a corpus-level POS analysis on the full LLaVA-665K dataset by merging subword tokens into words, computing word-level VIG scores, and applying POS tagging (using spaCy). The results show a clear hierarchy: proper nouns have the highest mean VIG (0.514), followed by nouns (0.144) and verbs (0.106), while function tokens such as particles (−0.002) and symbols (0.001) cluster near zero. Visually grounded content words consistently exhibit larger loss reductions, and function words contribute minimally. We will include this analysis in the revised appendix.

---

> > ### Author Rebuttal · Reviewer_g14S · 2026-04-03
> >
> > I recognize the authors' efforts to mitigate language bias through filtered, high visual impact training data; however, both the objective of eliminating language priors and the techniques employed are common ideas.
> > To enhance the paper's contribution, the authors should provide a more rigorous empirical analysis justifying why leveraging this technique during the training stage is superior to inference-time intervention methods, such as VCD. Specifically, the analysis in Section 4.4 should be expanded to include VCD , rather than only comparing the proposed method against the base model.

---

> > > ### Author Response · Authors · 2026-04-05
> > >
> > > We appreciate the reviewer's follow-up questions and the opportunity to clarify further.
> > >
> > > **Expanded analysis in Section 4.4**
> > >
> > > Following the reviewer's suggestion, we extended the analysis in Section 4.4 to include VCD.
> > > * Attention fraction (Fig. 4): This analysis measures how much the model itself attends to visual tokens, reflecting changes in its internal representations. Since VCD does not modify model parameters, comparing it here is not meaningful as the attention distribution is identical to the vanilla model by design. VIG training, by contrast, substantially increases attention to visual tokens across all layers, confirming that the model's visual grounding is strengthened at the representational level.
> > > * Text corruption robustness (Fig. 5):
> > > | Model | Base | Corruption | Norm |
> > > |---|---|---|---|
> > > | Baseline | 77.9 | 32.1 | 41.2 |
> > > | + VCD | 78.1 | 33.5 | 42.9 |
> > > | + VIG training | 78.2 | 42.3 | 54.1 |
> > > 	VCD yields only marginal improvement under text corruption (+1.4), as the model's internal reliance on visual input has not changed. VIG training produces a substantially larger gain (+10.2), confirming that it strengthens visual grounding at the representational level.
> > >
> > > **VIG vs. VCD: similarities and differences**
> > >
> > > Building on these results, we summarize the relationship between VIG and VCD.
> > >
> > > * Similarities:
> > > 	- Both compute the difference between outputs under original and degraded visual conditions.
> > > 	- Both aim to mitigate language bias in LVLMs.
> > >
> > > * Differences:
> > > 	- When: VCD is applied at every inference step. VIG computes it once before training.
> > > 	- How: VCD subtracts degraded-image logits during decoding. VIG uses the loss difference to select training samples and tokens.
> > > 	- Effect on model: VCD leaves the model unchanged. VIG shapes what the model learns from, improving visual grounding at the representational level.
> > > 	- Cost structure: VCD requires an extra forward pass per query at deployment. VIG adds a one-time scoring cost before training, with zero inference overhead.
> > >
> > > We argue that VIG is a more fundamental solution since it addresses language bias at its source in the training data, rather than correcting its symptoms at inference. Please refer to the evidence below:
> > > - VIG increases attention to visual tokens across all layers (Fig. 4); VCD does not change attention patterns at all.
> > > - Under text corruption, VIG improves robustness by +10.2 vs. VCD's +1.4 (Fig. 5).
> > > - VIG consistently improves all benchmarks without trade-offs (Tab. 3); VCD degrades some metrics (e.g., MMVet 28.62 → 27.01).
> > > - VIG and VCD produce additive gains when combined (Tab. 4), confirming that VIG resolves a layer of bias that VCD alone cannot reach.
> > >
> > > We hope this analysis addresses the reviewer's concern. We are grateful for the constructive feedback that helped strengthen the paper, and we will incorporate these results and discussions into the revised manuscript.

---

### Decision · Program_Chairs · 2026-04-30

**Decision:**

Accept (regular)

**Comment:**

The paper was positively received for its simplicity, clarity, and practical framing. Reviewers appreciated that the proposed method is straightforward, easy to follow, and potentially useful as a being simple.

A substantial part of the discussion centered on the concerns raised by Reviewer qcKu regarding novelty, experimental breadth, and comparison to related approaches. In the author response and follow-up discussion, the authors addressed most experimental concerns.

However, reviewers also shared the same concern that the novelty is incremental. While the method is simple, which is nothing bad, the core idea is viewed as conceptually close to existing works. Therefore, the submission appears to be borderline.